# Biomass-Based Chemical Looping Gasification: Overview and Recent Developments

Nhut Minh Nguyen [1,2,*], Falah Alobaid [1], Paul Dieringer [1] and Bernd Epple [1]

1 Institute for Energy Systems and Technology, Technische Universität Darmstadt, Otto-Berndt-Straße 2, 64287 Darmstadt, Germany; falah.alobaid@est.tu-darmstadt.de (F.A.); paul.dieringer@est.tu-darmstadt.de (P.D.); bernd.epple@est.tu-darmstadt.de (B.E.)
2 Department of Chemical Engineering, Can Tho University, 3/2 Street, Can Tho 90000, Vietnam
* Correspondence: nhut.nguyen@est.tu-darmstadt.de

**Abstract:** Biomass has emerged as one of the most promising renewable energy sources that can replace fossil fuels. Many researchers have carried out intensive research work on biomass gasification to evaluate its performance and feasibility to produce high-quality syngas. However, the process remains the problem of tar formation and low efficiency. Recently, novel approaches were developed for biomass utilization. Chemical looping gasification is considered a suitable pathway to produce valuable products from biomass among biomass conversion processes. This review paper provides a significant body of knowledge on the recent developments of the biomass-based chemical looping gasification process. The effects of process parameters have been discussed to provide important insights into the development of novel technology based on chemical looping. The state-of-the-art experimental and simulation/modeling studies and their fundamental assumptions are described in detail. In conclusion, the review paper highlights current research trends, identifying research gaps and opportunities for future applications of biomass-based chemical looping gasification process. The study aims to assist in understanding biomass-based chemical looping gasification and its development through recent research.

**Keywords:** biomass; gasification; chemical looping gasification; carbon dioxide capture; oxygen carriers; syngas

## 1. Introduction

The combustion of fossil fuels (coal, petroleum, and natural gas) contributes the largest share of greenhouse gas (GHG) emissions and currently, the mitigation of these emissions is one of the most challenging global issues. The Paris Agreement aims to limit the temperature increase to 1.5 °C above pre-industrial levels [1]. According to international energy outlook 2016 (IEO 2016), the total world energy consumption will predictably increase by 48% from 2012 to 2040 due to growth in non-OECD Asia (including China, India, Southeast Asia), the Middle East, parts of Africa, and America [2]. According to the IEO 2016 reference case, fossil fuels will present the greatest energy source in the world in 2040, accounting for 78% of total world energy consumption [2]. Although coal is the slowest-growing energy source with 0.6% annual growth, it still accounts for a large proportion of world energy consumption. This is due to the fact that coal is abundant and less expensive than natural gas or oil. However, the utilization of coal is related to the challenge of increasing the efficiency of thermal power plants and reducing emissions such as carbon dioxide that is the primary greenhouse gas and its increase in the atmosphere mainly causes global warming. Germany has set a key goal to achieve at least a 40% cut in GHG emissions by 2020 and 80–95% by 2050 compared to its 1990 levels [3]. To reach this target, the increased use of renewable energy will play a key role. By 2050, power generation in Germany must be almost entirely based on renewable energy sources. The

European Union (EU) has planned to achieve a 20% share of renewable energy in total energy consumption across its members by 2020 [4].

Biomass has been considered one of the most important primary and renewable energy resources for a renewable and sustainable energy future due to its carbon-neutral renewable and abundant quantity, which can give more carbon credit for conversion technologies and consequently economic advantages. Additionally, biomass has higher reactivity and higher volatile content, which can promote its conversion reactions. Its low content of sulfur and mercury leads to lower $SO_x$ emissions and pollutants. Other benefits of biomass are low ash content, thereby reducing solid residue, handling, and processing costs overall [5]. Combined with the carbon capture and storage processes, the overall system could achieve carbon negative emissions [6,7].

Biomass conversion pathways can typically be classified as biochemical and thermochemical processes. The biochemical conversion pathway consists of two main processes used, namely fermentation and anaerobic digestion [8]. Fermentation, a metabolic process, converts solid fuels into biofuels or chemicals through the action of enzymes, while anaerobic digestion is a conversion of organic material into biogas in the absence of oxygen. The biochemical processes have main disadvantages such as low energy efficiency, high water requirement, long conversion times, and stringent feedstock requirements [9]. The thermochemical conversion processes mainly include combustion, gasification, and pyrolysis. Biomass thermochemical conversion processes are characterized by low efficiency mainly due to biomass properties such as high moisture content and relatively low energy density. Biomass gasification, a thermochemical conversion approach, is to convert efficiently the solid fuels into a combustible gas mixture, mainly CO and $H_2$, which can be used as a feedstock in the production of chemicals or power generation. However, the drawback of conventional gasification technology is a demand for a large amount of heat supply for the production of high-quality syngas, making the process less attractive. Therefore, a new technology is required to be more economically feasible to produce enriched hydrogen syngas from biomass.

Ishida et al. [10] firstly proposed the term "chemical looping" for the process, where a metal oxide is used as an oxygen transport medium to perform a redox reaction scheme for an increase of exergy efficiency in power generation. In the chemical looping concept, oxygen carriers (e.g., $Me_xO_{y-1}/Me_xO_y$) [5–7] are applied for oxygen transport, avoiding direct contact between fuels and air. Chemical looping processes can be used for power generation, production of syngas, chemicals, and liquid fuels through chemical looping combustion (CLC) [9,11,12], chemical looping reforming (CLR) [13,14], and chemical looping gasification (CLG) [9,15]. In the CLC process, metal/metal oxide as oxygen carrier circulates between two reactors to completely combust fuels (gaseous and solid fuels), while the CLR is a process for the partial oxidation of hydrocarbon fuels to produce hydrogen. The CLG shares similar principles with the CLC and CLR, but the CLG can produce useful combustible gas from gaseous and solid carbonaceous materials through the partial oxidation process.

The typical mechanism operation of CLG of the biomass process is illustrated in Figure 1. The configuration mainly consists of an air reactor (AR) and a fuel reactor (FR), where oxidation and reduction reactions take place, respectively.

In the FR, a metal oxide as an oxygen carrier (OC) is reduced to provide oxygen for fuel conversion. Then, the reduced metal oxide is circulated to the AR to be re-oxidized before a new cycle. The general chemical reactions in the FR and AR are shown as follows.

$$\text{Biomass} + Me_xO_y \rightarrow Me_xO_{y-1} + H_2 + CO + CO_2 + CH_4 \tag{1}$$

$$Me_xO_{y-1} + \frac{1}{2}O_2 \rightarrow Me_xO_y \tag{2}$$

where $Me_xO_y$ is the oxidized and $Me_xO_{y-1}$ represents the reduced form of oxygen carrier [16].

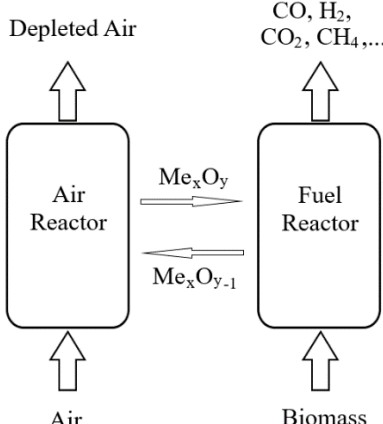

**Figure 1.** Schematic principles of Biomass chemical looping gasification process.

Biomass-derived chemical looping gasification is a novel technology to convert biomass into renewable hydrogen-enriched syngas. Since most reactions in the FR are endothermic, they require a large energy supply for stable operation. On the other hand, oxidation reactions in the AR are exothermic. Thus, the oxygen carrier, which can circulate between two reactors in the CLG process, can transfer not only oxygen but also heat from the AR to the FR through oxidation-reduction reactions [9,17,18]. Additionally, looping materials (metal oxides) can reduce considerably the amount of tar that causes serious problems in a biomass gasification system. Metal oxides act as an effective catalyst for tar cracking and reduction in tar formation during chemical looping gasification. Mendiara et al. [19] found that the tar content was reduced around 2.4% per degree Celsius with the presence of iron ore. The $NiFe_2O_4$ showed a dual-function of oxidation-catalyst for toluene reduction and significantly promotes toluene converted into carbon and $H_2$ [20]. Although biomass-based chemical looping gasification currently remains several challenges, the potential achievements may outweigh the challenges. Many studies have been carried out to investigate its nature and to solve operational challenges for the commercialization of this technology. The efforts focus on the development of biomass gasification processes for large-scale applications, improvement in reactivity and stability, as well as the multifunctional nature of looping materials, holistic evaluation for the economic feasibility of biomass-based chemical looping systems, and various types of biomass feedstock for chemical looping processes [9]. Therefore, biomass chemical looping technology has notable potentials to convert biomass-based materials into valuable products and effectively mitigate $CO_2$ emissions in the atmosphere.

Review papers that systematically analyzed scientific publications on certain topics are very valuable but are only rarely published. Although BCLG has been reviewed in previous works [9,14,15,18], the results and insights of recent studies of BCLG based on oxygen carriers have not been analyzed systematically. This work is to describe the novel gasification technology, which aims to elucidate the latest advances in chemical looping gasification. The objective of this work is to critically approach the research results on the chemical looping gasification technology of biomass and its progress through recent finding experimental and simulation/modeling studies to assist in the knowledge of the behavior and the potentials of using biomass in chemical looping gasification. The review paper is organized into six sections.

Section 2 is dedicated to the process configuration of BCLG, focusing on the key components of a CLG system such as system configurations, reactors. Furthermore, system complexity and its challenges are introduced in this section.

Section 3 presents an overview of looping materials in BCLG. In this section, fundamentals and developments of oxygen carriers used in BCLG are described according to recent studies.

Section 4 analyzes the influence of operating parameters on the performance of BCLG. This section may provide an understanding of the operation of BCLG under various conditions.

Section 5 summarizes the recent results and insights of BCLG through experimental and simulation works across the world. A large number of experimental studies have been developed to evaluate process performance in various operating conditions, system configurations, and types of oxygen carriers. These investigations can be categorized into kinetic studies and pilot-scale investigations. Additionally, simulation/modeling of the BCLG system is mainly developed for prediction, evaluation, and optimization. Their results can offer a good understanding of the feasibility of the commercialization of this process.

The paper closes with Section 6, highlighting the summary of this work and prospects of chemical looping gasification of biomass. Finally, the review paper concludes with concrete recommendations for this field of research.

## 2. Process of Chemical Looping Gasification of Biomass

Chemical looping gasification is a novel technology to convert biomass into gaseous products; it has been proven that its advantages, e.g., high-quality syngas production, lower $CO_2$ emissions. Furthermore, heat generated in the air reactor can be supplied to endothermic reactions in the fuel reactor through oxygen carriers, allowing for autothermal operation [17]. Additionally, the inorganic compounds in biomass ash can act as effective catalysts for gasification reactions [9], which is an important benefit for the chemical looping gasification of biomass.

### 2.1. Process Description

Chemical looping technology is designed to avoid direct contact between air and fuel by circulating metal/metal oxide acting as oxygen carrier between two reactors, i.e., air reactor (AR) and fuel reactor (FR). The metals/metal oxides work as oxygen carriers to transport oxygen from the air to the fuel via reduction-oxidation (redox) reactions. Biomass chemical looping gasification (BCLG) shares the principles with chemical looping technology. A simplistic mechanism of chemical looping gasification is illustrated in Figure 2. Biomass is partially oxidized in the FR by metal oxides ($Me_xO_y$) to produce a mixture of gases, mainly $H_2$, CO, and $CO_2$. Steam or $CO_2$ may be added to the FR to promote reforming reactions and char gasification reactions. A general description of the overall oxidation reaction in the FR is given in Reaction (4). In the AR, the reduced form of oxygen carriers ($Me_xO_{y-1}$) in the fuel reactor is oxidized by oxygen from the air as shown in Reaction (3). The key reactions in the fuel reactor are summarized in Table 1.

$$\text{Air reactor: } 2Me_xO_{y-1} + O_2 \rightarrow 2Me_xO_y \tag{3}$$

$$\text{Fuel reactor : } \text{Biomass} + Me_xO_y \xrightarrow{H_2O/CO_2} CO + H_2 + CO_2 + CH_4 + \text{tar} + Me_xO_{y-1} \tag{4}$$

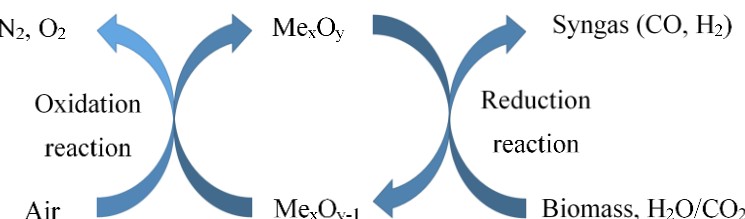

**Figure 2.** General scheme of biomass chemical looping gasification process.

**Table 1.** Chemical reactions in fuel reactor [19,21,22].

| No. | Name of Reaction | Chemical Reaction |
|---|---|---|
| 1 | Boudouard reaction | $C + CO_2 \leftrightarrow 2CO$ |
| 2 | Char reforming/water gas | $C + H_2O \leftrightarrow CO + H_2$ |
| 3 | Methanation | $C + 2H_2 \leftrightarrow CH_4$ |
| 4 | Water gas shift reaction | $CO + H_2O \leftrightarrow CO_2 + H_2$ |
| 5 | Steam reforming of methane | $CH_4 + H_2O \leftrightarrow CO + 3H_2$ |
| 6 | Dry reforming | $CH_4 + CO_2 \leftrightarrow 2CO + 2H_2$ |
| 7 | Oxygen carrier reduction | $CO + Me_xO_y \rightarrow CO_2 + Me_xO_{y-1}$ |
| 8 | Oxygen carrier reduction | $H_2 + Me_xO_y \rightarrow H_2O + Me_xO_{y-1}$ |
| 9 | Oxygen carrier reduction | $CH_4 + Me_xO_y \rightarrow 2H_2 + CO + Me_xO_{y-1}$ |
| 10 | Oxygen carrier reduction | $C + Me_xO_y \rightarrow CO + Me_xO_{y-1}$ |
| 11 | Oxygen carrier reduction | $C + 2Me_xO_y \rightarrow CO_2 + 2Me_xO_{y-1}$ |
| 12 | Tars' reforming | $Tars + H_2O \leftrightarrow CO + H_2 + CO_2 + hydrocabons + \ldots$ |
| 13 | Hydrocarbon reforming | $Hydrocabons + H_2O \leftrightarrow CO + H_2 + CO2 + \ldots$ |

### 2.2. Process Classification

Sharing the basic principles with chemical looping technology, the BCLG process takes place based on many intrinsic components such as the fuel, gasifying agents, reactor configurations, and looping materials. These parameters can be grouped into three main categories as shown in Figure 3. In the following, each of these categories is elucidated in detail.

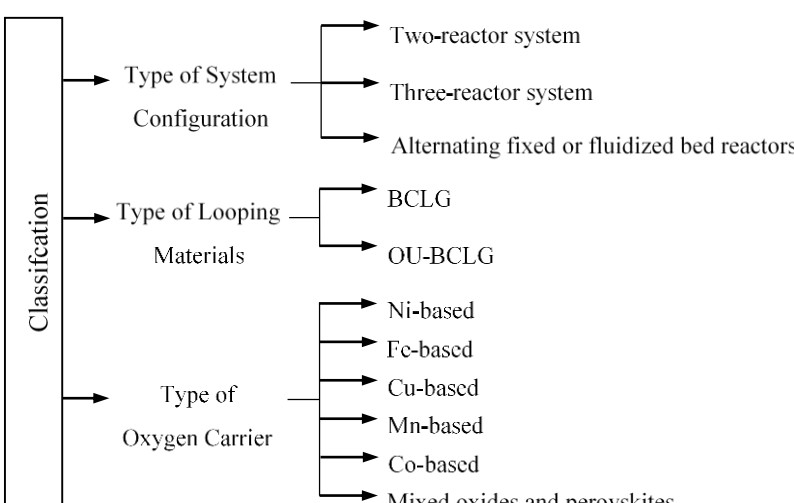

**Figure 3.** Classification of biomass chemical looping gasification process.

2.2.1. System Configuration

The biomass chemical looping gasification concept is to produce high-quality syngas for further applications. The contact between the fuel and the oxygen carrier plays a key role in the chemical looping system, especially BCLG. Hence, the selection of the reactor configuration is an important criterion for chemical looping processes. The essential requirements for selecting an appropriate BCLG with a continuous operation are as follows [12,17,23]:

(i)   There should be sufficient particle circulation between the FR and the AR.
(ii)  There should be sufficient contact between the fuel/air and the solid oxygen carriers to achieve maximum conversion.
(iii) High temperature and high-pressure operations must be carried out to achieve higher overall efficiency.
(iv)  There should be limited gas leakage between the FR and the AR.

As a part of a chemical looping system, a reactor is a crucial factor that affects process performance. Two common types of reactors have been proposed for chemical looping

applications, namely fixed-bed and fluidized-bed reactors. Fixed-bed reactors are the simplest type of reactor in chemical looping processes ranging from laboratory-scale to pilot plant-scale and commercial-scale. In this type of reactor, the solid materials are stationary and are alternately exposed to reducing and oxidizing conditions through periodic switching of feed streams [24]. The major advantages of the fixed-bed reactor are that separation of gas and solid particles is not required, which allows for better utilization of oxygen carrier. To achieve a high process energy efficiency and continuous operation, two or more fixed bed reactors in parallel can be installed in the system. However, this reactor configuration has not been used widely for BCLG since it shows heat and mass transfer limitations and demands high temperature and a complex flow switching system. In fluidized bed systems, solids behave like a fluid by passing gas or liquid upwards through the bed of particles. The fluidized-bed reactor is extensively used in chemical looping processes. Its advantages over the fixed-bed reactors are uniform temperature distribution, more effective mixing, and higher heat and mass transfer. The behavior of a fluidized bed strongly depends on flow gas velocity and solid properties. Among fluidization regimes, bubbling, turbulent, and fast fluidization are mainly applied in chemical looping processes. However, one of the serious problems in the stable operation of the fluidized-bed reactor is particle segregation leading to poor fluidization. Based on the key requirements and types of reactors in the system mentioned above, it could be proposed to be accomplished in three configurations like a two-reactor system, three-reactor system, and alternating packed or fluidized bed reactor (Figure 4). Many researchers have carried out their studies in different types of reactor systems and different types of oxygen carriers for CLG fitting biomass fuels to evaluate the feasibility of the BCLG process.

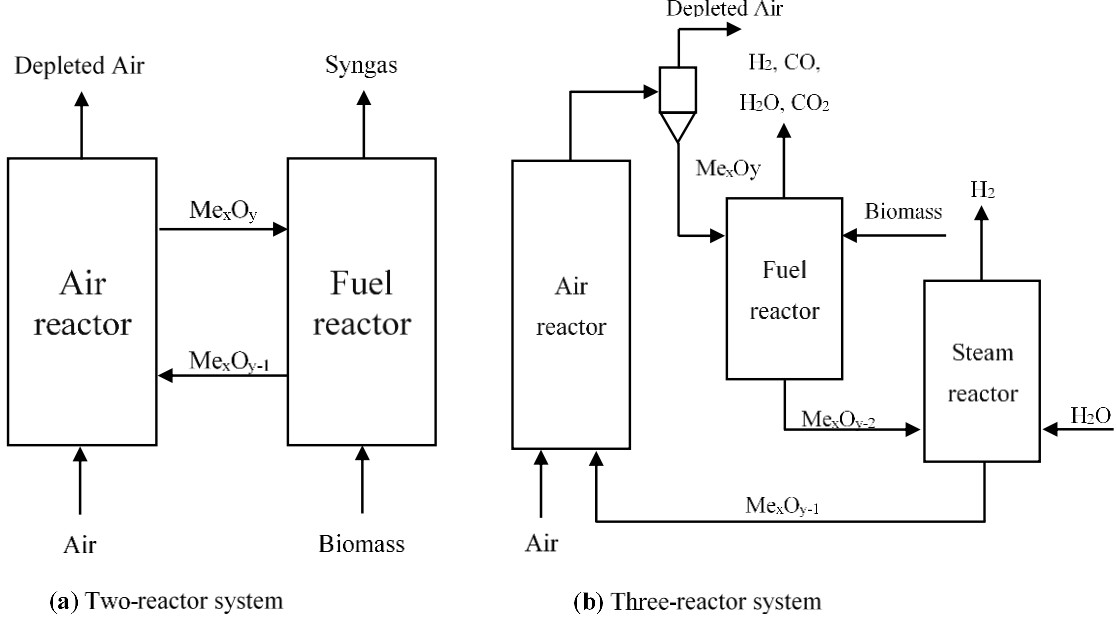

**Figure 4.** BCLG configurations for hydrogen enrich gas production: (**a**) Two-reactor system and (**b**) Three-reactor system.

Two-Reactor System

The two-reactor system is the most popular configuration for BCLG as shown in Figure 4a [25–28]. The configuration consists of two fluidized bed reactors as the AR and FR, respectively. In the AR, the oxygen carrier materials are oxidized by oxygen from air. The oxidized form of the oxygen carrier is transferred to the FR to react with biomass to produce a gaseous mixture and the reduced form of metal oxides, then they are returned to the AR for regeneration. Additionally, two loop-seal devices are installed between the AR and FR to prevent gas mixing between two reactors. This configuration is based on the reactivity of the oxygen carrier considering that the residence time of the

oxygen carrier required for the reduction reaction is higher than for the oxidation. The AR, a fast fluidized bed reactor, has two objectives: to give the driving force for the solid material circulation and provide sufficient oxygen and heat for fuel conversion in the FR [11]. Interconnected fluid fluidized bed reactors, a type of two-reactor system, comprise mainly two fluidized bed reactors. This configuration normally consists of a high-velocity riser and a low velocity bubbling fluidized-bed as the AR and FR, respectively, being the most popular configuration among all the various types [26,28–33]. Biomass gasification takes place in the FR while the oxygen carrier is oxidized inside the AR. The loop-seals are installed to prevent gas leakage between the AR and the FR. Additionally, cyclones are used to remove solid particles from the gas stream.

Three-Reactor System

The three-reactor system can generate pure hydrogen and syngas separately and simultaneously. It shares similar principles with the chemical looping reforming of methane configuration [14] as shown in Figure 4b. A simplistic mechanism of the three-reactor system for BCLG combined with water splitting is illustrated in Figure 5. Biomass is partially oxidized by the lattice oxygen of the metal oxides in the FR to produce syngas as shown in Reaction (4), but the reduced oxygen carrier is oxidized by steam to regenerate lattice oxygen and produce $H_2$ in the steam reactor Reaction (5) instead of in the AR. Afterward, the oxygen carrier is fully oxidized in the AR Reaction (3) before continuing the next cycle.

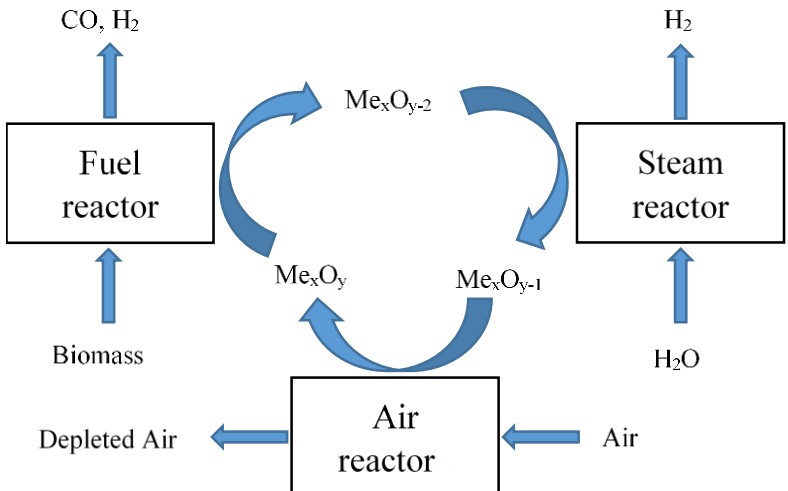

**Figure 5.** Process scheme of three-reactor system for BCLG.

The reaction in the steam reactor can be illustrated as follows:

$$\text{Steam reactor: } Me_xO_{y-2} + H_2O \rightarrow H_2 + Me_xO_{y-1} \tag{5}$$

where $Me_xO_y$ is an oxygen carrier, $Me_xO_{y-1}$ and $Me_xO_{y-2}$ are the corresponding reduced form of oxygen carriers with different reduction degrees, i.e., the strongly reduced oxygen carriers ($Me_xO_{y-2}$) leaving the FR are partially oxidized in the steam reactor ($Me_xO_{y-1}$) before full regeneration in the AR.

Oxygen carrier material used in the process demands a sufficiently high reactivity with biomass, good performance for water splitting to generate hydrogen, and high stability during redox cycles. Additionally, the material should have good resistance to carbon deposition because it may cause contamination of the hydrogen produced. Some metal oxides have been considered as possible oxygen carriers for this configuration such as $Fe_3O_4$, $WO_3$, $SnO_2$, Ni-ferrites, (Zn, Mn)-ferrites, Cu-ferrites, and Ce based oxides [14]. A study was developed in a fixed bed reactor by He et al. [34] to combine BLCG and water/$CO_2$ splitting using $NiFe_2O_4$ as oxygen carrier. In this study, experimental investigations were

carried out separately in three steps. Firstly, biomass was reduced by $NiFe_2O_4$ to generate syngas in the presence of steam/$CO_2$. Afterward, the reduced form of oxygen carrier was oxidized partially by steam/$CO_2$ to produce $H_2$/CO, then it was fully oxidized by air in the oxidation step. During the investigations, syngas and $H_2$/CO were obtained separately in different steps. The authors also proposed phase transitions corresponding to different reduction degree of oxygen carrier at different steps as follows:

$$NiFe_2O_4 \rightarrow metallic\ Fe(Ni)/FeO_x \rightarrow Ni_{1.43}Fe_{1.7}O_4/Ni \rightarrow NiFe_2O_4$$

The three-reactor configuration has been considered a promising approach since it can produce syngas and pure hydrogen simultaneously. However, there have been very few studies on this configuration. Some research related to the three-reactor system has been focused on gaseous fuels, coal, and $FeO$/$Fe_3O_4$/$Fe_2O_3$ materials as looing materials [35–41].

Packed and Fluidized-Bed

The packed bed configuration reactor can be applied for chemical looping gasification of biomass. A simple configuration of this type is shown in Figure 6. The system comprises at least two reactors in parallel working alternately to continuously produce syngas. Each reactor works alternately reduction and oxidation cycles and intermittently alternated with short periods of mild fluidization of the bed after each cycle to level off temperature and concentration profiles [12]. The main advantages of this technology include the separation of gas and particles and the ability to work under high pressure, whereas this technology requires high temperature and a high flow gas switching system [11].

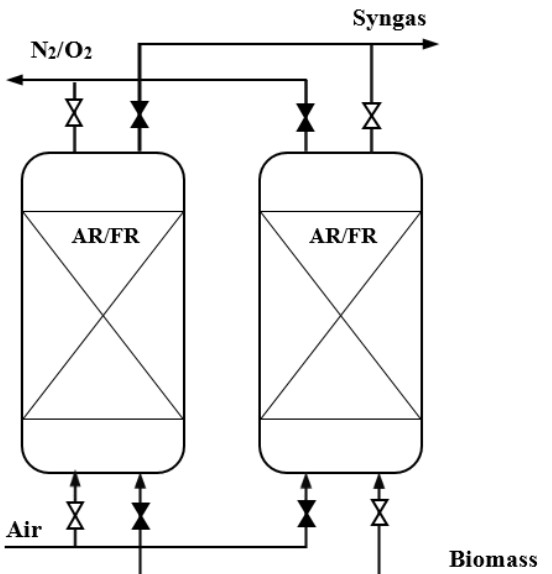

**Figure 6.** Simple configuration of alternating packed or fluidized bed reactors.

Yan et al. [42] used a fixed bed reactor to study the performance of $Al_2O_3$/$BaFe_2O_4$ as a synthesized oxygen carrier in BCLG for hydrogen-rich syngas production. Liu et al. [43] conducted a study of chemical looping co-gasification of pinewood and polyethylene in a fixed reactor. The effects of operating conditions during the BCLG process were reported in the investigation. A study of CLG of biomass char using NiO-modified iron ore as an oxygen carrier was carried out by Huang et al. [44]. They reported the reduction of oxygen carrier by biomass char in TGA and a fixed-bed reactor. Liu et al. [45] developed $Ca_2Fe_2O_5$ with Mg/Al/Zn oxides as support materials for BCLG in a fixed bed reaction. This work mainly investigated the effects of Mg/Al/Zn oxides on the reactivity of $Ca_2Fe_2O_5$ and the BCLG performance. Wang et al. [46] presented experimental results in a bubbling fluidized bed reactor for CLG of sawdust pellet with high volatile and low ash content as fuel. It was

found that higher reaction temperatures increased gas production, while the amount of liquid and solid decreased.

### 2.2.2. System Complexity and Challenges

Although BCLG has many potential advantages over the conventional biomass gasification process, this technology has remained several challenges for widely commercialized applications. Thus, these issues should be considered and solved for its commercialization.

#### System Complexity

Biomass owns intrinsic characteristics, such as high moisture content, low energy density, various geometrical shapes, ash content, etc. Furthermore, different types of biomass have different properties [47,48]. The bulk density of lignocellulosic biomass materials is relatively low (80–150 kg/m$^3$ for grass biomass and 160–220 kg/m$^3$ for woody biomass), which can cause low efficiency in transportation and storage. Therefore, pre-treatment processes are required to homogenize its properties and meet the requirements of chemical looping gasification. These processes possibly include drying, densification, or grinding, etc. These processes require energy and cost resulting in reducing process efficiency.

The BCLG generally operates at high temperatures (over 800 °C) for the production of high calorific value syngas. The process requires a large heat transfer area in a gasifier for a sufficient heat supply for biomass gasification, which increases the operating complexity of the system.

During biomass gasification, biomass ash is produced, which would result in reducing the performance of oxygen carriers. The removal of looping materials from biomass ash (including unreacted carbon) is a considerable challenge to minimize the loss of the looping materials. Additionally, interactions between the ash and looping materials may contribute to a reduction in separation efficiency [9]. Moreover, the syngas produced from the gasification system also contains unwanted substances such as dust, fly ash, tar, alkali metals, nitrogen compounds, sulfur compounds, chlorine, and trace elements. Thus, the CLG system should be integrated with the syngas cleaning system to remove these impurities from syngas before further applications. The required level of syngas cleaning significantly depends on the end-use technology and/or emission standards [49]. There are several methods to remove contaminants which are conveniently categorized according to the process temperature range as hot gas cleanup (HGC) (>300 °C), cold gas cleanup (CGC) (<100 °C), and warm gas cleanup (WGC). The majority of these methods are developed based on using wet scrubbers [49,50].

Typically, biomass gasification system efficiency for power generation would increase corresponding to an increase in unit capacity. However, most of the power plants from biomass are small-scaled and decentralized due to difficulty in the biomass supply chain [51].

The stable load on the chemical plant and long-term operation are among the challenges. Load variations or frequent start-ups and shut-downs could result in damage to the equipment and a reduction in productivity. Moreover, it could cause an increase in cost due to excessive fuel consumption during the start-up and shut-down.

#### Fouling and Corrosion

Tars are produced during the pyrolysis stage of a gasification process. Certain high molecular weight tar compounds, which start to condense at lower than 450 °C and subsequent polymerization, can cause clogging and blockages in gas lines and filters and reduce process efficiency [52]. Furthermore, the removal of the condensed tars consumes intensive energy and cost. Ash is the remained inorganic residue and exists as particulate after biomass chemical looping processes at temperatures of 800–1200 °C. The particulate of the fuel reactor can deposit in downstream equipment and gas lines resulting in blockages and attrition; however, these particles can be removed by using filters and cyclone separators [5]. Another concern is that biomass fuels have a high content of minerals, including sodium,

potassium, phosphorous, and chloride which can cause potential corrosion for equipment, particularly heat transfer surfaces. Several researchers investigated that chloride has a catalytic effect that promotes the dissociation of the steel material in the heat exchangers, even at low temperatures (100 to 150 °C) [53]. Corrosion mechanisms coexist and occur simultaneously, including several chemical reactions between the metal and metal oxides with gaseous substances of Cl and O, the solid phase of alkali metal salts (K and Na), and ash in the liquid phase and during phase changes [54]. Furthermore, the high content of silica from many types of biomass sources [48] and the formation of molten material in the ash can also cause significant abrasion and erosion on mechanical components and pneumatic ash handling systems [53]. Additionally, biomass-derived tars also have a highly acidic nature (typical pH < 2) and high water content; thus, its presence can cause considerably corrosion-related problems in downstream pipelines and equipment of the system [52].

## 3. Looping Materials in Chemical Looping Gasification of Biomass

### 3.1. Looping Materials

In the BCLG process, oxygen carriers are used to transfer lattice oxygen from the air to the fuel via redox reactions to produce syngas, which can avoid the direct contact of air and fuel. Metal oxide materials play a key role in the chemical looping redox processes. These metal oxide materials can be classified as chemical looping gasification (BCLG) and oxygen uncoupling chemical looping gasification (OU-BCLG) based on the properties of looping materials as shown in Figure 3. As can be seen in Figure 7a, biomass fuel is decomposed and cracked down at high temperatures mainly into three products of gas, tar, and char, which can be simplified into volatiles and char. The volatile matter in the biomass, which is reduced to the release of hydrocarbon gases, is constituted by complex organic substances and can be condensed at a sufficiently low temperature to liquid tars. The gaseous fraction is an incondensable mixture of gases at ambient temperature and accounts typically for 70–90 wt.% of the feedstock [55,56]. This mixture of gases consists mainly of hydrogen, carbon monoxide, carbon dioxide, and light hydrocarbon. Afterward, these products react with oxygen carrier particles. Two main types of reactions occur simultaneously in the fuel reactor: homogeneous and heterogeneous. There are two reaction pathways proposed between the oxygen carrier and biomass in the fuel reactor: direct reduction of oxygen carrier by biomass and reduction of oxygen carrier by gaseous biomass gasification products [9]. The first pathway consists of two main reaction types, reactions of volatile matter released from biomass with oxygen carrier (see Reaction (7)), and direct solid-solid reactions. Since solid–solid reactions are very limited due to low contact efficiency, the impact on the final gas composition is negligible in comparison to heterogeneous reactions. Due to a relatively high fraction of volatile matter in biomass, it is a benefit for biomass that a higher proportion of the fuel can directly react with oxygen carrier in a CLG system [9].

The second pathway is an indirect reaction between biomass and oxygen carrier. Firstly, Biomass is gasified with $H_2O/CO_2$ to produce mainly $H_2/CO$ (see Reactions (8) and (9)), and then gaseous products can readily react with oxygen carrier (see Reactions (6), (10) and (11)).

The general reactions occur in the fuel reactor:

$$\text{Biomass} \rightarrow \text{Volatiles} + \text{Char (mainly C)} + H_2O \tag{6}$$

$$\text{Volatiles} + Me_xO_y \rightarrow CO_2 + H_2O + Me_xO_{y-1} \tag{7}$$

$$\text{Char} + CO_2 \rightarrow 2CO \tag{8}$$

$$\text{Char} + H_2O \rightarrow H_2 + CO \tag{9}$$

$$CO + Me_xO_y \rightarrow CO_2 + Me_xO_{y-1} \tag{10}$$

$$H_2 + Me_xO_y \rightarrow H_2O + Me_xO_{y-1} \tag{11}$$

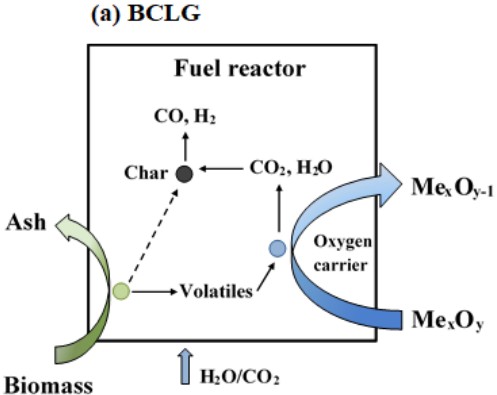

**Figure 7.** Main reactions in the fuel reactor: (**a**) BCLG and (**b**) OU-BCLG.

The other concept proposed is oxygen uncoupling biomass chemical looping gasification (OU-BCLG). The system is based on the use of an oxygen carrier that can release gaseous oxygen in the fuel reactor to oxidize the fuel as shown in Figure 7b.

$$2Me_xO_y \rightleftharpoons 2Me_xO_{y-1} + O_2 \tag{12}$$

$$Char + O_2 \rightarrow CO + CO_2 \tag{13}$$

$$Volatiles + O_2 \rightarrow CO + CO_2 + H_2O \tag{14}$$

In the OU-BCLG process, the oxygen carrier releases oxygen according to Reaction (12), and the biomass fuel can be decomposed simultaneously into volatiles and char as shown in Reaction (6). Afterward, the devolatilization products are oxidized to CO, $CO_2$, and $H_2O$ according to Reactions (13) and (14). The remaining char is gasified by $CO_2$ and $H_2O$ to produce CO and $H_2$. Moreover, the fuel reactor in both concepts should be fluidized by $H_2O$, $CO_2$, or their mixture, which also acts as gasifying agents to accelerate biomass gasification.

There are a limited number of oxygen carrier materials, which have the property of releasing oxygen, that can meet the requirement for multiple cycles of oxygen uncoupling processes. They must be reversible in the reactions of releasing and oxidizing oxygen. In comparison with oxygen carrier for the normal BCLG process, the metal oxides used in OU-BCLG have a suitable equilibrium partial pressure of gas-phase oxygen at a temperature range of 800–1200 °C. Thus, there are three metal oxide system could be used in the OU-BCLG system such as $CuO/Cu_2O$, $Mn_2O_3/Mn_3O_4$, and $Co_3O_4/CoO$. Their reversible reactions are as follows [11]:

$$4CuO \rightleftharpoons 2Cu_2O + O_2 \ \Delta H_{850} = 263.2 \text{ kJ/mol } O_2 \tag{15}$$

$$6Mn_2O_3 \rightleftharpoons 4Mn_3O_4 + O_2 \ \Delta H_{850} = 193.9 \text{ kJ/mol } O_2 \tag{16}$$

$$2Co_3O_4 \rightleftharpoons 6CoO + O_2 \ \Delta H_{850} = 408.2 \ kJ/mol \ O_2 \tag{17}$$

Unlike the chemical looping combustion process, the BCLG concept is to produce useful combustible gas (CO and $H_2$, syngas); therefore, the partial oxidation of biomass can be achieved by using oxygen carrier materials suitable for OU-BCLG. Due to the very high reactivity of oxygen uncoupling materials, less metal oxide material used is needed in the system which could reduce the reactor size and associated costs. OU-BCLG is a new concept and a few studies related to OU-CLG have been performed with biomass [57–59].

### 3.2. Type of Oxygen Carrier

Oxygen carrier plays a key role as the chemical intermediate to indirectly transfer pure oxygen from the air to the fuels via redox reactions in the chemical looping processes. In chemical looping gasification, an oxygen carrier is used to not only provide the oxygen needed for gasification to extremely improve the quality of syngas but also as a thermal carrier that increases heat balance between the two reactors [27]. Furthermore, some metal oxide oxygen carriers may have a catalytic effect on biomass tar cracking [60–62]. Thus, the selection of an appropriate oxygen carrier is one of the most important criteria for the good performance of the chemical looping process. The preferable properties of an oxygen carrier for system performance should be as follows [11,12,63–65]:

(i)     Sufficient oxygen transport capacity.
(ii)    Favorable thermodynamics and reactivity regarding the solid fuel for the reduction reactions.
(iii)   High reactivity in the oxidation reactions.
(iv)    Selectivity towards CO and $H_2$.
(v)     Resistance to attrition to minimize losses of elutriated solid.
(vi)    Minimal carbon deposition.
(vii)   Good fluidization characteristics (no presence of agglomeration) and high melting points.
(viii)  Reasonable cyclability/circulation for using over several redox reactions.
(ix)    Low cost and long lifetime.
(x)     Environmentally friendly properties.
(xi)    High mechanical strength and resistance to frictional stresses.
(xii)   Capability of converting biomass to gaseous products.
(xiii)  Propensity to convert methane

In the chemical looping systems, the lattice oxygen in metal oxides can oxidize partially or fully carbonaceous fuels; therefore, the reduction behavior of the metal oxides significantly affects the performance of chemical looping gasification systems. A modified Ellingham diagram is shown in Figure 8a, which depicts the standard Gibbs free energies of reactions as a function of temperature according to the following reaction:

$$aMe_xO_y + O_2 \rightleftharpoons aMe_xO_{(y+2/a)} \tag{18}$$

The diagram shows the change of the Gibbs free energy with temperature variation, which can be used to evaluate the redox potentials of common oxygen carrier materials. According to the thermodynamic analysis, metal oxides can be mainly categorized in three zones according to their potential to oxidize the fuel or desired applications. In Figure 8b, three zones are bound by three reaction lines as follows:

$$\text{Reaction line 1: } 2CO + O_2 \rightarrow 2CO_2 \tag{19}$$

$$\text{Reaction line 2: } 2H_2 + O_2 \rightarrow 2H_2O \tag{20}$$

$$\text{Reaction line 3: } 2C + O_2 \rightarrow 2CO \tag{21}$$

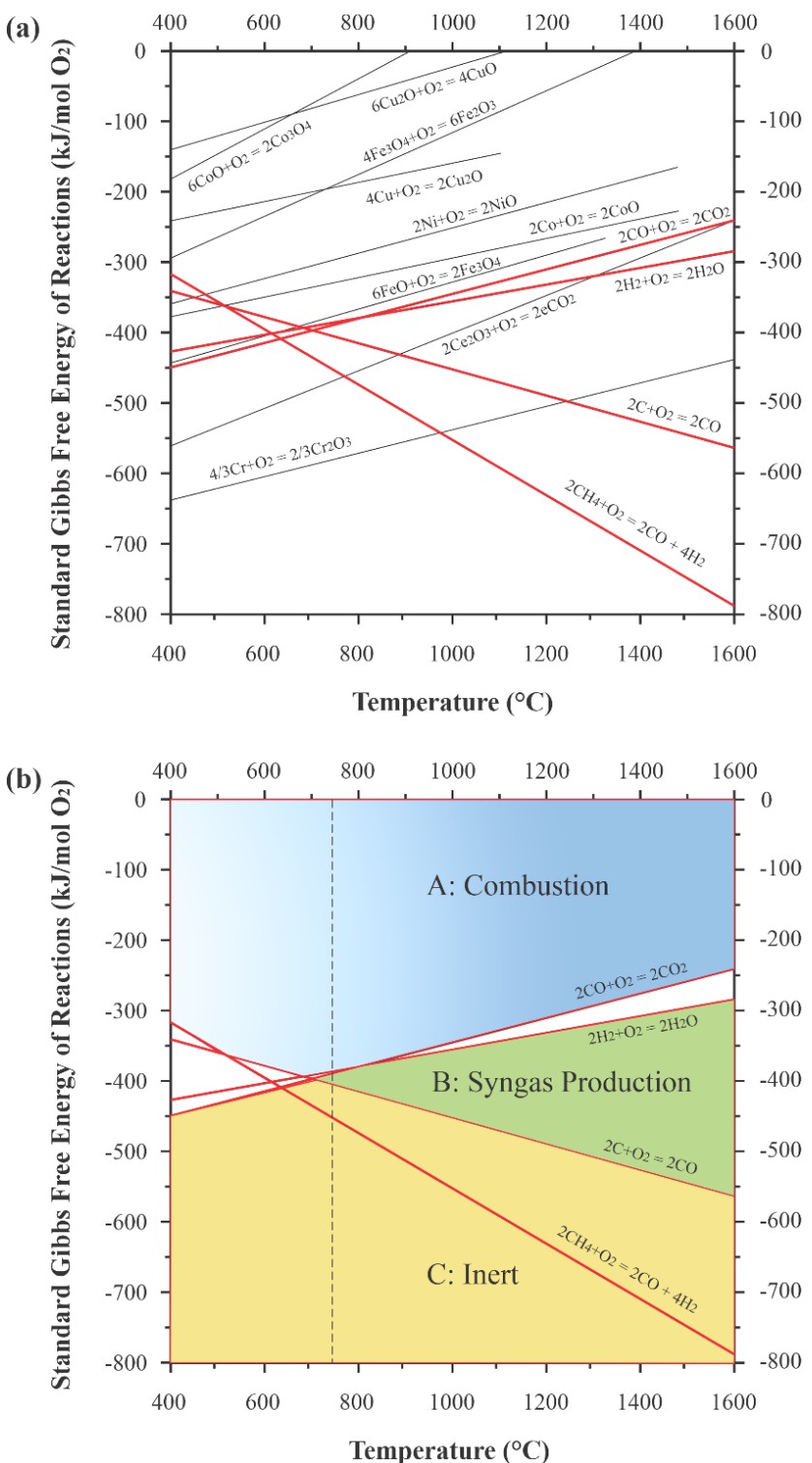

**Figure 8.** Modified Ellingham diagram for oxygen carrier material: (**a**) standard Gibbs free energy changes of some oxidation reactions and (**b**) reaction zones for syngas production applications [9].

Zone A: metal oxides located in the area above lines 1 and 2 have strong oxidizing properties and can completely or near completely convert the fuel to yield 95% of $CO_2$ [66]. Metal oxides in this zone consist of $CuO$, $Co_3O_4$, $Fe_2O_3$, $Cu_2O$, $NiO$, $CoO$, and $Fe_3O_4$.

Zone B: the region is bound in between lines 1 and 3, the materials in this region can partially oxidize the fuel to produce $CO$ or $H_2$ and an excessive quantity of oxygen

carrier does not yield full oxidation. Thus, metal oxides in zone B are theoretically ideal for chemical-looping gasification applications with 90% of syngas yield [66].

Zone C: the materials in this zone below line 3 cannot be used as oxygen carriers due to their low oxidizing ability; thus, they are considered inert for chemical looping applications.

Ellingham diagrams allow effective evaluation of redox pairs thermodynamically, which can provide theoretical indications for the oxygen carrier's selection. An oxygen carrier performs in a chemical looping system depending on a combination of reaction kinetics, a ratio of oxygen carrier to fuel, contact time, and process design.

To select oxygen carriers for chemical-looping gasification applications, there are two approaches to achieve partial oxidation of fuels [9]. Firstly, metal oxides in zone B are used to mainly produce CO and $H_2$, which cannot be completely oxidized due to thermodynamic restrictions. The other approach is to use under stoichiometric quantities of materials in zone A.

According to the basis of desired characteristics, various types of oxygen carriers have been investigated for chemical looping gasification of biomass including Fe-based [26,32,43,67–70], Ni-based [25,44], Cu-based [71,72], Mn-based [28], Zn [73], bimetallic oxygen carriers (Fe-Ni [31,74], Fe-Cu [71,75], etc.). They can be classified as follows:

(i)　　Ni-based oxygen carriers.
(ii)　　Cu-based oxygen carriers.
(iii)　Fe-based oxygen carriers.
(iv)　Mn-based oxygen carriers.
(v)　　Co-based oxygen carriers.
(vi)　Perovskite-type complex metal oxides
(vii)　Other oxygen carriers.

Several kinds of research have been conducted to compare a large number of oxygen carriers. These results show the general reactivity of the commonly used metal oxides follows orders [9,14,76]:

$$NiO > CuO > Mn_3O_4 > Fe_2O_3 \tag{22}$$

One of important characteristics of an oxygen carrier is oxygen transport capacity, $R_O$, which is defined as the usable oxygen in the oxygen carrier during one redox cycle, defined by

$$R_O = \frac{m_{ox} - m_{red}}{m_{ox}} \tag{23}$$

where $m_{ox}$ and $m_{red}$ represent the weight of fully oxidized and reduced oxygen carrier in the redox cycle, respectively. The oxygen transport capacity, $R_O$, determines the fuel conversion and the amount of solid circulation rate.

Table 2 presents the $R_O$ values of different redox couples of oxygen carriers. It can be seen that $CaSO_4$, $Co_3O_4$, NiO, and CuO have higher $R_O$ values, whereas reduction of $Fe_2O_3$ to $Fe_3O_4$ produces only around 3 wt.% of oxygen, the lowest one in the list. The support materials are commonly used to improve the oxygen carrier's performance. The interaction of the metal oxide and the support material can influence the $R_O$ value of oxygen carriers. By using $Al_2O_3$ supported Fe-based oxygen carriers, more $Fe_2O_3$ can be reduced to FeO in form of $FeAl_2O_4$ with $R_O$ of 0.045, allowing it to almost completely combust to $CO_2$ and $H_2O$ in a chemical looping system.

**Table 2.** Oxygen transport capacity of some Redox couples, $R_O$ [11].

| Redox Couples | $CaSO_4/CaS$ | $Co_3O_4/Co$ | $CuO/Cu$ | $CuO/Cu_2O$ | $NiO/Ni$ |
|---|---|---|---|---|---|
| $R_o$ | 0.47 | 0.27 | 0.20 | 0.1 | 0.21 |
| Redox couples | $Mn_2O_3/MnO$ | $Mn_2O_3/Mn_3O_4$ | $Fe_2O_3/FeO$ | $Fe_2O_3/Fe_3O_4$ | $Fe_2TiO_5/FeTiO_3$ |
| $R_o$ | 0.1 | 0.034 | 0.1 | 0.034 | 0.05 |
| Redox couples | $CuAl_2O_4/Cu.Al_2O_3$ | $CuAlO_2/Cu.Al_2O_3$ | $CuAl_2O_4/Cu.AlO_2$ | $Fe_2O_3/Al_2O_3/FeAl_2O_4$ | $FeAl_2O_4/Ni.Al_2O_3$ |
| $R_o$ | 0.089 | 0.066 | 0.044 | 0.045 | 0.091 |

Among oxygen carriers, the Nikel-based oxygen carriers perform the best reactivity of the reduction and oxidation reactions in chemical looping gasification of biomass [25,31,34]. Due to high oxygen transport capacity, very high chemical reactivity, and almost complete conversion of hydrocarbons, NiO/Ni should be utilized in sub-stoichiometric quantities for chemical looping gasification applications. Additionally, Ni-based oxygen carriers exhibit the potentials for chemical looping process applications at high temperatures in the range of 900–1100 °C, but significant challenges include accumulative chemical and thermal stress, and mechanical degradation with high cycle number [77] along with low reaction rate, high cost, and sulfur deactivation have hindered for their commercial applications.

Copper-based oxygen carriers perform good properties in the chemical-looping process, e.g., high reactivity, oxygen transport capacity, cyclical, and mechanical stability [78]. Reactions of CuO and the gasification products are exothermic thus could promote biomass reforming [71,75]. However, the main challenges of Cu-based oxygen carriers are sintering and de-fluidization due to the low melting temperature of metallic Cu.

The oxygen transport capacity of Fe-based oxygen carriers is relatively low, and it cannot be reduced further than $Fe_3O_4$ in a fluidized bed reactor because of the thermodynamic limitation [25]. Thus, to improve the reaction rate and characteristics of metal oxides, alkali, and alkaline compounds are added to the oxygen carrier. Yu et al. [79] investigated that the addition of alkali metals to $Fe_2O_3$ can improve the reaction rate of solid fuel, and Gu et al. [80] reported that $K_2CO_3$ -added iron ore showed stable catalysis properties for coal chemical looping combustion, whereas calcium oxide exhibited not only catalytic activity but also the ability to capture $CO_2$ and sulfur compounds [81]. The known benefits of iron oxide are low price, chemical stability, non-toxicity, low sintering temperature needed, and low degree de-fluidization problems, the Fe-based materials are promising oxygen carriers in chemical looping processes. Fe-based oxygen carriers perform differently in their reactivity with various fuels, their reaction rate varies with the following order: $H_2$ > CO > $CH_4$ > solid fuels [69,82].

Mn-based oxygen carriers are probably classified into the group of oxygen uncoupling materials with $Mn_2O_3/Mn_3O_4$ phases, but $Mn_2O_3$ is not stable at relevant temperatures; therefore, $Mn_3O_4/MnO$ can be the main phase transition in the chemical looping processes which is not possible to release oxygen [28]. Mn-based materials show low-cost and environmentally friendly properties that are similar to Fe-based oxygen carriers. MgO addition improved the oxygen release ability of the oxygen carrier and increased the performance of biomass chemical looping gasification [45]. The theoretical oxygen transport capacity of Mn-based oxygen carriers is higher than that of Fe-based materials, but there are a few reports that have been published regarding these materials in the chemical looping gasification of biomass.

Perovskite-type complex metal oxides have a general formula $ABO_3$, where A is a lanthanide ion and/or alkaline earth metal and B is a transition metal ion [18]. Many studies pointed out that two different oxygen species being in the perovskite oxides, such surface absorbed oxygen and bulk lattice oxygen which play a different role in chemical looping processes [83]. The surface absorbed oxygen combusts completely methane, whereas the other is responsible for partial oxidation of methane to $H_2$ and CO. Redox activity of the

perovskite materials is dependent on the mixed-conductivity of the support. Many types of research found that the materials released between 3.44 and 8.23 wt.% of oxygen at 600 °C [18]. Oxygen transport within iron oxide particles can play an important role at the final stage of the reduction, whereas higher $Ni^{2+}$ shows a greater reduction ability. The advantages of the materials are excellent regeneration ability and less agglomeration at temperatures above 100 °C, as well as high thermal stability, good mechanical properties, and high selectivity to synthesis gas.

Along with transition metal oxides, i.e., Ni, Cu, Co, Mn, Fe have been investigated as possible oxygen carriers for the chemical looping gasification process, synthetic oxygen carriers are widely researched, while natural minerals are potentially developed due to their low cost. The low-cost natural minerals are used as oxygen carriers including iron ore, ilmenite, manganese ore, and waste materials from the steel industry and alumina production [63]. Synthetic materials generally comprise single metal oxides, as well as their blend with inert support ($Al_2O_3$, $SiO_2$, $MgAl_2O_4$, $TiO_2$, $ZrO_2$, CaO, etc.) [11] to improve their reactivity, mechanical properties, and stability, as well as multifunctional properties, including catalytic function, tar decomposition, and $CO_2$ adsorbent in BCLG. Many studies have been carried out to investigate experimentally the reactivity and stability of different oxygen carriers with various supporting materials, e.g., $Al_2O_3$ [25,26,33,42,45,78,84], $SiO_2$ [30], CaO [43,85,86], MgO [45], $TiO_2$ [69,84]. It was found that the reactivity and stability of oxygen carriers increase significantly during the multiple cycles when supported with $Al_2O_3$. Additionally, $TiO_2$ is used as a supporting material with iron oxides for chemical looping processes. The compounds performed high reactivity and stability in various studies [32,69,84]. Ilmenite performed well in the continuous operations of a 1.5 $kW_{th}$ pilot reactor with >90% of biomass conversion and carbon conversion efficiency obtained [32]. It was reported that the presence of MgO and $ZrO_2$ can increase thermal and chemical stability, as well as specific heat capacity of oxygen carriers [45,87]. The addition of CaO to oxygen carrier has emerged in the chemical looping gasification process for continuous hydrogen production [43,45,85,88–90]. Calcium oxide (CaO) has been known as a catalyst and $CO_2$ sorbent, which favors hydrogen-rich gas production [43,85,91]. Furthermore, the presence of CaO also works as tar cracking catalyst for hydrogen production in a fluidized bed reactor [92,93]. High hydrogen content (approximately 63%) and hydrogen yield (23.07 mol/kg rice straw) at 800 °C were obtained with the presence of calcium-iron oxide oxygen carriers [86], while Sun et al. [93] reported the maximum hydrogen yield reaches 7.12 mol/kg pine wood at 850 °C.

Many kinds of research have been carried out by mixing different active metal oxides or mixing different oxygen carriers composed of single metal oxides. The major aims of using mixed oxides are desired as follows [11,31,34,78,94,95]:

(i)     Increase the reactivity and/or stability of particles.
(ii)    Improve the conversion of the fuel gas.
(iii)   Improve the mechanical strength and resistance to the attrition of particles.
(iv)    Improve tar decomposition.
(v)     Improve carbon dioxide adsorption.
(vi)    Decrease the carbon deposition.
(vii)   Decrease the preparation cost of the oxygen carrier.
(viii)  Decrease the use of toxic metals.

Many studies have been carried out to evaluate the performance of many metal oxides as oxygen carrier for hydrogen production. It was found that the Fe-based oxygen carrier shows the most attractive application prospect for the chemical looping applications, particularly for hydrogen production due to its high-temperature stability, low cost, environmentally friendly effects [14,96]. Composite materials of iron oxide and perovskite have been proposed as a potential oxygen carrier material for hydrogen production due to their excellent oxygen transport properties and stability under cycling. Thus, these materials work effectively for catalytic oxidation reactions including hydrogenation, CO oxidation, and catalytic combustion [14]. According to Dueso et al. [97], several oxygen

carrier materials composed of LSF731 and iron oxide showed higher performances in oxygen capacity and stability, as well as hydrogen production compared to iron oxide. The characteristics of common types of oxygen carriers are summarized in Table 3.

**Table 3.** Overview of common types of oxygen carriers in chemical looping processes [9,11,12,14,18].

| Type of OC | Operating Temp. [°C] | Support Materials | Advantage | Disadvantage |
|---|---|---|---|---|
| Ni-based | 900–1100 | $Al_2O_3$, $MgAl_2O_3$, $ZrO_2$, Bentonite, $TiO_2$, MgO, $SiO_2$ | Very high reactivity and selectivity, strong catalytic properties for hydrocarbon conversion, high oxygen transport capacity, high stability, low agglomeration | Sulfur deactivation, high cost, health, and safety issues |
| Fe-based | | $Al_2O_3$, $MgAl_2O_4$, $TiO_2$, $SiO_2$, YSZ, $CeO_2$, $ZrO_2$ | Environmentally friendly and non-toxicity, low cost, high mechanical strength, high chemical stability | Relatively low reactivity, low oxygen transport capacity, agglomeration issue, low solid circulation rate |
| Cu-based | <800 | $Al_2O_3$, $CuAl_2O_4$, $TiO_2$, $SiO_2$, $CeO_2$, $ZrO_2$, Bentonite, MgO, $MgAl_2O_4$ | High reactivity and oxygen transport capacity, low cost and toxicity, high chemical and mechanical stability, environmentally friendly. No demand for external heat | Agglomeration and de-fluidization due to the low melting point of Cu |
| Mn-based | | $Al_2O_3$, $MgAl_2O_4$, $TiO_2$, $SiO_2$, $ZrO_2$, bentonite | Low cost and non-toxicity, environmentally friendly | Sulfur deactivation, relatively low oxygen transport capacity, agglomeration, low reactivity with fuels |
| Co-based | | YSZ, $Al_2O_3$, $CoAl_2O_4$, $TiO_2$, $SiO_2$, $ZrO_2$, bentonite | High oxygen transport capacity, high reactivity with $CH_4$ and CO | High cost and environmental impact and health issue, low reactivity with fuels |

### 3.3. Performance of Looping Materials

Oxygen carrier is a key factor in chemical looping processes. One of the main challenges of these processes is to stabilize their performance over prolonged redox cycling. However, many reasons may reduce the performance of oxygen carriers during Biomass chemical looping gasification. As discussed above, the agglomeration of oxygen carriers is one of the most serious challenges in chemical looping processes which can lead to bed defluidization and deactivation of oxygen carriers. The Cu metal has a low melting point which can cause agglomeration at high temperatures, and the redox rate of CuO would readily decrease after few reduction-oxidation cycles [98], while several agglomerations of Fe-based oxygen carrier has been reported when the phase change to $Fe_3O_4$ during oxidized in air. Therefore, different support materials have been used to mitigate this problem. Another concern is the reduction of the mechanical strength of oxygen carriers during chemical looping processes. This problem can lead to attrition and decrease their lifetime. Along with physical attrition, chemical stress during redox cycles is also a reason for the reduction of mechanical strength [9].

In solid–gas reactions, carbon deposition on the solid oxygen carriers is a serious problem, because it can reduce their reactivity and shorten their lifetime. Carbon deposition is a complex phenomenon that is affected by several factors (pressure, temperature, availability of oxygen in the oxygen carrier, contents of water vapor, and $CO_2$ in the fuel gas) [99]. For Ni- or Cu-based oxygen carriers, carbon deposition was reported when the oxygen

carrier converted more than 75% [9,100]. Additionally, gas sulfur compounds in the syngas possibly react with oxygen carriers and produce various sulfur compounds that may cause the deactivation of looping materials. During thermochemical conversion processes, the inorganic matters in biomass are converted into the solid residual produced that can cause several problems for the performance of oxygen carriers and the system. Biomass ash composition is a complex mixture of inorganic-organic matters with solid, liquid, and gaseous phases from various origins. In the solid phase, ash is composed of inert materials from feedstock and un-reacted matters. In chemical looping processes, the interaction between biomass ash and oxygen carrier could be a significant issue. Gu et al. [101] reported that the performance of iron ore oxygen carrier was significantly influenced by the biomass ash type, resulting in serious particle sintering and subsequent deactivation of oxygen carrier with $SiO_2$-rich ash, whereas K-rich ash with lower $SiO_2$ content could efficiently improve the performance of oxygen carrier during cycle experiments. Additionally, biomass ash has considerably low melting temperatures (<800 °C), which is mostly lower than the operating temperatures of the gasification process ranging between 800 and 1000 °C. Consequently, it can cause agglomeration, sintering, and defluidization problems, as well as ash deposition [9,30,102,103].

## 4. Effect of Process Parameters on Biomass Chemical Looping Gasification

Operating conditions play a key role that significantly influences the process performance, especially the amount of hydrogen in the product gas. The effects of the key process parameters in BCLG are described and analyzed in this section.

### 4.1. Biomass Characteristics

Biomass is a complex mixture of organic and inorganic substances. Vassile et al. [48] categorized types of biomass based on their biology diversity, source, and origins, such as woody plant, herbaceous and agricultural plants/grasses, aquatic plants, human and animal waste, contaminated and industrial waste biomass, and biomass mixtures. Florin and Harris [104] characterized biomass as follows:

(i)    The chemical constituents: cellulose, hemicellulose, and lignin.
(ii)    Elemental composition.
(iii)    Inherent mineral content.
(iv)    Proximate analysis: moisture, volatile, fixed carbon, and ash contents.
(v)    Physical properties: particle size, shape, and density.

All these parameters have been investigated their effects on the product gas composition, yield, and process performance. The properties of different types of biomass are shown in Tables 4 and 5. Many studies have been carried out on various types of biomass for gasification and BCLG. Generally, the higher content of cellulose and lignin produces more gaseous products resulting in the increasing potential of hydrogen recovery from biomass. The hydrogen production from biomass gasification is attributed to the intrinsic properties, moisture content, and alkali content [105]. Additionally, biomass with high contents of oxygen and hydrogen results in highly liquid and volatile yields, thereby reducing the overall energy conversion efficiency of the combustion process. Furthermore, higher H/C ratios in fuels generate a greater heat of combustion, whereas higher ratios of O/C produce more $CO_2$ emission per amount of energy release [47,106]. A simplified summary of biomass pyrolysis is illustrated in Figure 9 [107]. Some studies [108–111] found that biomass with the proportions of cellulose and hemicellulose are directly related to higher CO and $CO_2$ concentrations, while higher lignin content leads to a higher char yield and $CH_4$ concentration during the pyrolysis process. Consequently, those intermediate products can vary in the final composition of the product gas. Chang et al. [112] carried out an experimental study on biomass gasification for hydrogen production in a fluidized bed reactor.

**Table 4.** Biomass constituents (wt.% on a dry basis) [21,29,47].

| Biomass Type | Cellulose | Hemicellulose | Lignin |
|---|---|---|---|
| Hardwood | 42–50 | 20–38 | 16–25 |
| Softwood | 35–50 | 24–35 | 16–33 |
| Straws | 33–40 | 20–45 | 15–20 |
| Corn stover | 33–35 | 21–24 | 17–22 |
| Switchgrass | 30–50 | 10–40 | 5–20 |

**Table 5.** Proximate analysis of some biomass feedstock (wt.%) [32,33,47,49].

| Biomass | Moisture [a] | VM | FC | Ash | LHV (MJ/kg) |
|---|---|---|---|---|---|
| Wood | 20 | 82 | 17 | 1 | 18.6 |
| Wheat straw | 16 | 59 | 21 | 4 | 17.3 |
| Barley straw | 30 | 46 | 18 | 6 | 16.1 |
| Torrefied woodchips | 5.28 | 70.75 | 22.82 | 1.15 | 19.3 |
| Pine sawdust | - | 84.1 | 15.6 | 0.4 | 19.3 |
| Pine wood | 5.6 | 78.5 | 15.3 | 0.6 | 17.4 |

[a] Intrinsic.

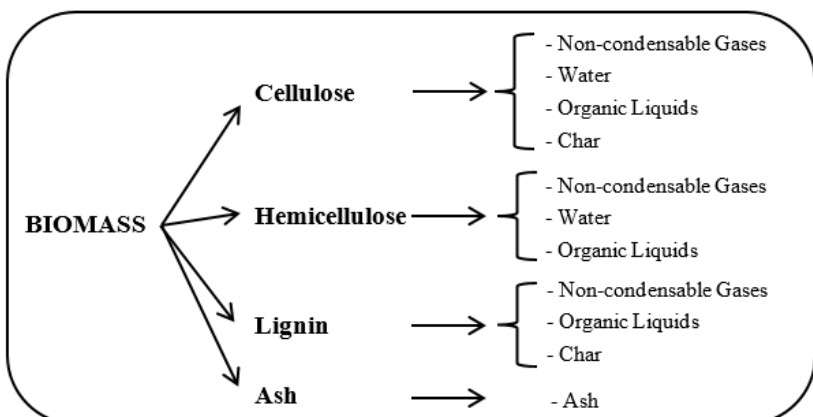

**Figure 9.** The simplified summary of biomass pyrolysis.

They gasified three different types of biomass ($\alpha$-cellulose, bagasse, and mushroom waste) at different temperatures (600–1000 °C), steam-to-biomass ratios, and equivalent ratios. The results showed that the highest hydrogen production was obtained in the case of using $\alpha$-cellulose, which has higher contents of carbon and hydrogen than those of the other species. Raut et al. [113] studied the effect of torrefaction pretreatment on biomass gasification performance at different reaction temperatures (700–850 °C) and the steam-to-biomass ratio (0.6). In the study, biomass feedstock was Poplar wood and its torrefied products. The hydrogen content increased corresponding to the increment of torrefaction degrees.

Biomass particle size influences significantly the performance of biomass gasification and hydrogen production. Biomass with smaller particles provides a larger surface area per unit mass resulting in improving heat and mass transfer which promotes the gasification reactions (Boudouard reaction, water gas reaction) to produce significantly $H_2$ and CO. Several studies have been conducted to analyze the effect of biomass particle size on gasification reactions. Lv et al. [114] found that $CH_4$, CO, and $C_2H_4$ were produced more in the case of the smaller particles while less $CO_2$ was formed, and larger particles showed greater heat transfer resistance which led to incomplete pyrolysis and a higher amount of unreacted char. Hernández et al. [115] reported that the smaller particle size of the fuel improves mass and heat transfer due to higher external surface area/volume resulting in

more porous owing to a higher amount of volatile release. Therefore, the reactivity of the remaining char increases leading to an increase in the gasification reactions. Di Blasi [116] in his work demonstrated that the smaller particles with a large surface area produce more light gases, as well as less unreacted char and tar. Thus, the particle size of biomass has a significant impact on the product gas yield.

*4.2. Gasification Temperature*

Temperature is one of the most significant factors in gasification since the gasification process is a thermochemical conversion process that uses heat to convert the fuel into product gas. Generally, most of the biomass gasification reactions are endothermic reactions, an increase in temperature promotes them. Additionally, higher temperatures increase the heating rate among the particles resulting in effective destruction of the particles and proceeds for complete gasification reactions [22]. Consequently, more yield of gaseous products is generated, and the amount of unreacted char reduces.

Many researchers have studied the relationship between gasification temperature and process performance, as well as hydrogen production. As shown in Figure 10, hydrogen content slightly decreases in most cases, whereas studies conducted by Zeng [27] and He [34] show a small increase in hydrogen fraction in the product gas. The possible explanation supporting this trend can be referred to Le Chatelier's principle. Elevated temperatures favor the reactants in exothermic reactions and the products in endothermic reactions. As a result, the endothermic reactions in gasification would be strengthened with an increase in temperature which leads to an increase in the content of $H_2$ and CO, as well as more solid char is consumed to produce gaseous products. Furthermore, higher temperature also promotes cracking heavier hydrocarbon and tars.

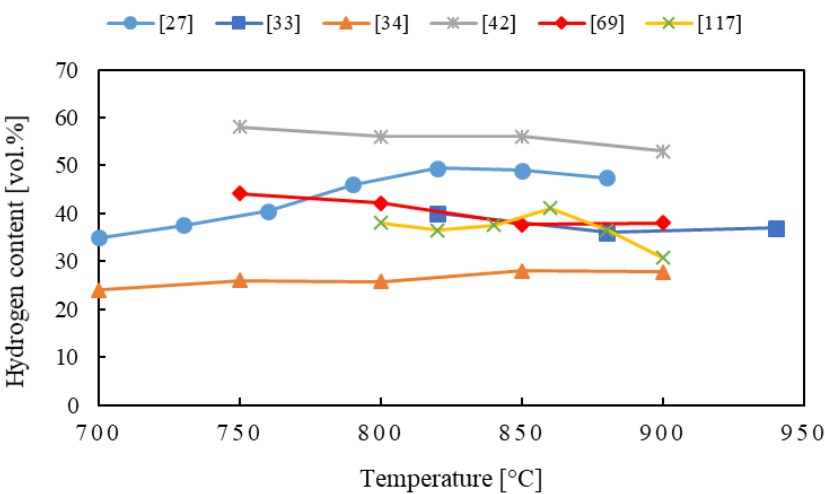

**Figure 10.** Hydrogen concentration as a function of fuel reactor temperature.

Additionally, elevated temperature improves gas yield and process efficiencies during the BCLG process. Yan et al. [42] reported that the total gas yield increased significantly with the gasification temperature. Additionally, the concentration of CO and $CH_4$ increased, while there was a slight decrease in the yield of $H_2$ and $CO_2$. Zeng et al. [27] investigated that $H_2$ content increased to the maximum value of 49.47% at 820 °C, then slightly dropped to approximately 47% at 880 °C. They also found that CGE increased with increasing temperature and reached a peak at 820 °C. Ge et al. [117] carried out a study on BCLG in a 25 kW$_{th}$ reactor using natural hematite as oxygen carrier. The temperature range in the study was between 800 and 900 °C. The results showed that hydrogen concentration reached the maximum value at 860 °C, then it decreased considerably to 30.77% at 900 °C. Gas yield showed a similar trend with hydrogen content, while carbon conversion efficiency increased with elevated temperatures. In a study of BCLG using $NiFe_2O_4$ [34], carbon conversion efficiency and syngas yield increased from approximately 26% and

$0.32 \text{ m}^3/\text{kg}_{\text{biomass}}$ from 700 to 850 °C. These variations of gas concentrations and total gas yield are related to a series of competing reactions in the gasifier. Under high temperatures, biomass pyrolysis, heavier hydrocarbon, and tars are promoted increasing gas yield and gas concentrations. High temperatures also thermodynamically favor the reactants in exothermic reactions and the products in endothermic reactions. At high temperatures, the reactivity of oxygen carriers with combustible gases is promoted, resulting in a reduction in the content and yield of hydrogen. This is rarely stated in small-scale experimental studies, but larger FR temperatures also mean that a higher heat transport (i.e., solid circulation) is required in the CLG unit.

### 4.3. Steam-to-Biomass Ratio

Steam-to-biomass ratio (SBR) refers to the amount of steam the mass of biomass fed into the gasifier. SBR is a key parameter strongly affecting hydrogen production and carbon conversion efficiency, as well as total gas yield. High SBRs result in a lower amount of unreacted char and greater both the yield and concentration of hydrogen in the product gas. Many researchers have reported that increasing SBR leads to a rise in hydrogen production and carbon conversion efficiency, as well as a low amount of tar produced. Figure 11 shows that hydrogen content is proportional to the ratio of steam-to-biomass.

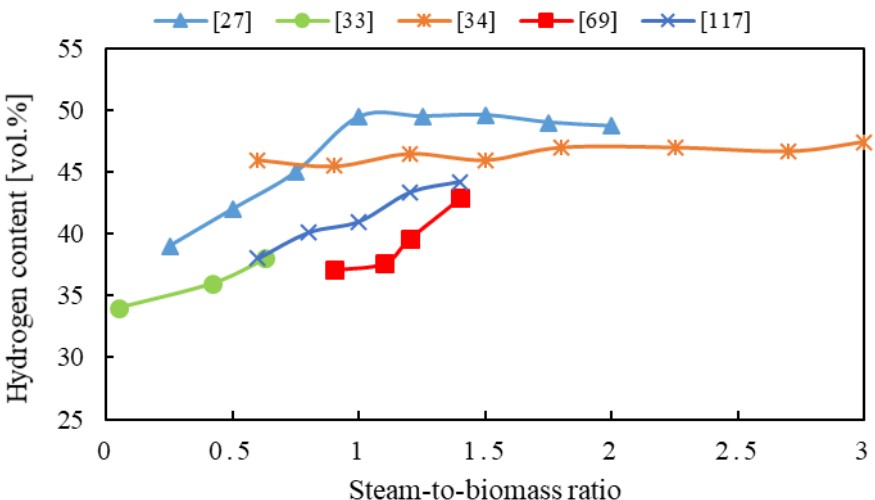

**Figure 11.** Hydrogen concentration as a function of steam-to-biomass ratio.

In the presence of high steam content in the gasifier, there is an increase in both the yield and content of hydrogen. Along with the increase in hydrogen production, process efficiencies have been improved corresponding to higher SBR. It should be noted that CO content decreases with higher SBRs. These tendencies are probably attributed to water gas shift reaction, char gasification reaction and steam reforming reaction. However, the undesired rise in $CO_2$ content has been reported which could be problematic in hydrogen-enriched gas production. This fact can be evidenced from study results [27,34,42,114,117,118].

Along with the enhancement of hydrogen production, the higher amount of steam used in the gasifier promotes tar decomposition, which is one of the most serious issues in biomass gasification. It is likely attributed to tar cracking reactions with the presence of steam. Reed [119] found that steam could strengthen participation of tar in steam gasification which leads to the decrease in tar content at higher SBR. Consequently, hydrogen and total gas yields can enhance as products of tar cracking reaction (tar reforming reaction). The addition of steam mainly brings about favorable aspects. Yet, in a large-scale process steam addition is one of the main sources of energy consumption. Hence, one would always try to minimize steam addition to the lowest possible level.

### 4.4. Oxygen Carrier-to-Biomass Ratio (OBR)

In the CLG process, metal/metal oxide provides lattice oxygen as a gasifying agent for solid fuel gasification. Oxygen carrier-to-biomass ratio (OBR) is defined as the mass ratio of the amount of oxygen carrier per biomass fed into the gasifier being a key factor in a BCLG process. OBR strongly influences the composition of the product gas, gas yields, and the performance of the BCLG process. Many studies have been carried out to evaluate the effect of oxygen carrier (in both type and quantity used) on the performance of BCLG, especially syngas production. It is noted that an increase in the amount of oxygen carrier used leads to further char gasification; therefore, high OBRs are beneficial for carbon conversion efficiency. However, it was found that a rise in OBR cause low content and yield of $H_2$ and CO in the product gas. It is due to the higher quantity of oxygen carriers that would promote the oxidation reactions of combustible gases including $H_2$ and CO. On the other hand, a high amount of oxygen carrier could result in a higher gasification temperature, which can enhance the gasification and improve the quality of the product gas to some extent.

Ge et al. [117] reported that a high amount of oxygen carrier could keep gasification temperature stable under the experimental conditions. Additionally, carbon conversion efficiency increased corresponding to the increase in the hematite mass percentage from 40 to 60 wt.%. However, $H_2$ and CO concentrations declined significantly by around 9% and 5%, respectively, resulting in the reduction in the syngas yield from 0.74 to 0.52 $Nm^3/kg$. An experimental study on BCLG using $BaFe_2O_4/Al_2O_3$ as an oxygen carrier [42] figured out that CO reduced with the increase of OBR for 30AF, whereas it increased for 30ABF at OBR of 0.6, then dropped down at higher OBRs. Huang et al. [67] used iron ore as an oxygen carrier to oxidize biomass char in a fixed bed reactor. They evaluated the effect of iron ore excess number, $\Omega$, which refers to the ratio of oxygen provided by oxygen carrier to the oxygen required for the complete oxidation of the fuel, on the performance of char gasification. Carbon conversion efficiency increased by approximately 23%, but $H_2$ and CO contents reduced by 7% and 10%, respectively, when the OBR increased from 0.46 to 1.17. Huijun et al. [25] showed an increase in carbon conversion efficiency while large amounts of $H_2$ and CO could be consumed with the rise in NiO content resulting in a significant decrease in syngas yield. An interesting result was found in a study on chemical looping co-gasification of biomass and polyethylene [43]. It was noted that there was an opposite variation of the contents of $H_2$ and CO with the increment of OBR. The concentration of $H_2$ increased from 34.78 vol.% to 38.15 vol.%, whereas CO and $CH_4$ contents declined with increasing OBR from 0 to 1.0. Additionally, more syngas yield also was generated with the elevated OBR. It can be considered that syngas production could be promoted by the presence of the oxygen carrier.

### 4.5. Effect of Operating Conditions on Tar Formation

Tar is a product of the gasification process which is defined as any matter in the product stream that is organic condensable compounds produced in thermochemical reactions [52]. Due to the high content of volatile matters in biomass, a large amount of tar derived from biomass gasification may cause many serious problems for gasifier or downstream processing steps as blockages, clogged filters, and efficiency reduction, but there are few investigations into its formation and effects in biomass chemical looping gasification.

High temperatures can thermally crack tar components efficiently, but they may reduce process efficiency. It was observed that naphthalene reduces by 47% when temperature increases from 820 to 940 °C [33]. Huang et al. [120] reported that the tar content declines from 18.58 to 9.03 $g/Nm^3$ in the range of operating temperature between 740 and 940 °C. Another study found that about 86% of tar content is eliminated when temperature increases from 500 to 900 °C [121].

Furthermore, the addition of steam in the gasifier can also abate an amount of tar in the product gas through steam reforming reactions of tar. Some publications found that steam

can effectively crack down large tar molecules [15,32,33]. Increasing the SBR would promote cracking reactions. However, this promotion can be less effective at a relatively high value of SBR. Moreover, excess amount of steam causes a temperature reduction in the reactor, resulting in low tar decomposition by temperature effect. Condori et al. [32] investigated the variation of the amount of tar formed as a function of the steam-to-biomass ratio. They found that an increase in the steam-to-biomass ratio results in a decrease in the amount of tar produced with a reduction of approximately 27% at 820 °C when the ratio increases from 0.06 to 0.9. A downward trend of tar yield was observed with a reduction of 73% when steam addition was further increased to the steam-to-biomass of 1.25 in an experimental investigation of biomass chemical looping gasification with $Cu_5Fe_5$ at 800 °C [121].

Figure 12 presents the tar yields of biomass chemical looping gasification under different gasification conditions and various oxygen carriers. The feedstocks used in these investigations were pine wood and pine wood sawdust. Without an oxygen carrier, the highest amount of tar was produced with about 22 and 17 $g/Nm^3$. In the presence of an oxygen carrier as catalyst and oxidation agent, tar yield obtained reduces significantly to the lowest yield of around 2 $g/Nm^3$. A similar trend was observed that tars were obtained between 0.9 and 3.0 $g/Nm^3$ at different operating conditions with $Fe_2O_3/Al_2O_3$ as oxygen carrier [33]. It is noteworthy that increasing the oxygen carrier-to-biomass ratio promotes the tar decomposition, but it can decline the reduction degree of the oxygen carrier, resulting in the low catalytic reactivity of the oxygen carrier and reduction in tar removal efficiency. Tian et al. [71] investigated the effect of oxygen carrier-to-biomass ratio on the reforming of tar using $Cu_5Fe_5$ as oxygen carrier. They found that the total amount of tar obtained reduces strongly by approximately 67% when the oxygen carrier-to-biomass ratio increases from 0 to 0.8. The results also perform that the presence of oxygen carrier can promote the transformation of macromolecular components into smaller molecular species.

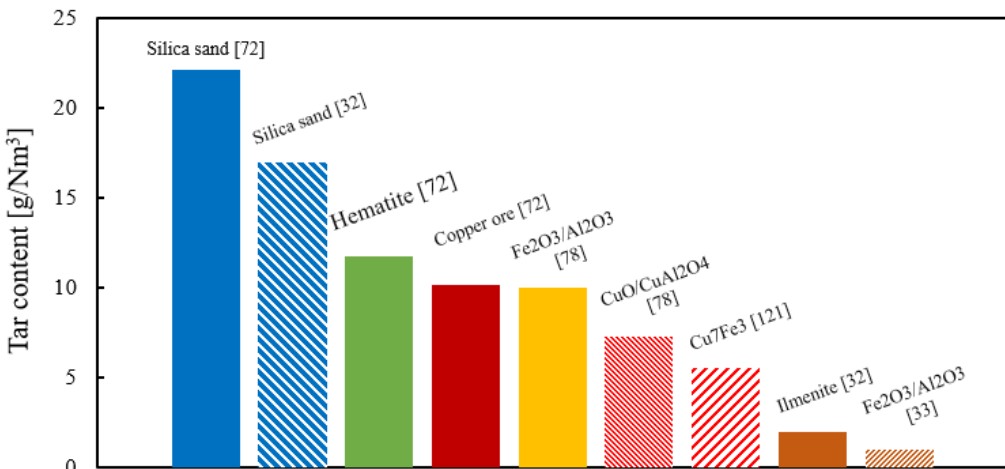

**Figure 12.** Tar yield (800–880 °C).

Hence, operating conditions have significant influences on tar formation, including its yield and constituents. Increasing operating parameters, i.e., gasification temperature, steam-to-biomass ratio, and oxygen carrier-to-biomass ratio, can be beneficial for tar removal from biomass chemical looping gasification.

## 5. Progress in Biomass Chemical Looping Gasification

Chemical looping gasification is viewed as a promising gasification technology for high-quality syngas generation. Currently, biomass is emerging as a renewable feedstock for the CLG process. Recent studies have been carried out to analyze kinetics and the process performance in pilot-scale systems, as well as to simulate the process through technical software. However, the research activities on chemical looping gasification of biomass are not

still limited compared to coal by this time, resulting in the lack of good understanding of the process behavior and hindering further large-scale commercial applications.

## 5.1. Thermodynamic Analysis and Kinetic Studies

Thermodynamic and kinetic studies are to evaluate the energy efficiency of the system and the effects of operation parameters on process performance and the interaction between solid fuel and oxygen carriers, as well as the feasibility of oxygen carrier for chemical looping gasification of biomass. The results obtained will be analyzed to develop pilot-scale and then industrial processes by mostly using thermogravimetric analysis equipment (TGA). Various types of oxygen carriers and their compounds have been investigated to assess their feasibility for the chemical looping gasification process. Many studies have been carried out to analyze the reactivity of an oxygen carrier in various atmospheres and to propose kinetic models to describe the kinetics of the oxygen carrier with different types of fuels [94,96,122–127]. During biomass chemical looping gasification, many gas-solid reactions take place simultaneously in the reactor. Therefore, the knowledge of redox reaction kinetics and their mechanism is necessary for the design of the system [124,125,128].

Yan et al. [122] analyzed system performances of biomass-coal co-gasification with steam. In the study, a coal and biomass-based chemical looping power generation system was developed. Wheat straw and coal were introduced into the reactor to gasify with steam, then the generated syngas would reduce $Fe_2O_3$ in the reducer. FeO generated from the reducer was oxidized by steam to produce hydrogen, and $Fe_3O_4$ generated in the oxidizer would be regenerated to form $Fe_2O_3$ before the next cycle. Hydrogen obtained in the oxidizer then was used as feedstock to generate electric power through the SOFC system. The result of the study was used to analyze the efficiency of energy and exergy of the system, the effect of operation parameters on system performance, and economic feasibility for the removal of $NO_x$, $SO_x$, and $CO_2$. Wang et al. [125] discussed the mechanism of reactions between corn cob and $Mn_2O_3$ during the chemical looping gasification. Furthermore, the method of Gibbs free energy minimization was used to analyze the thermodynamics of syngas generation from the process. They proposed the optimal ratio of $Mn_2O_3$/corn cob is 0.18 to obtain the maximum yields of $H_2$ and CO, and the total dry concentration of CO and $H_2$ could gain 98.8% at 1000 °C and atmospheric pressure. Additionally, the carbon conversion also increases in the presence of steam used as a gasifying agent.

Non-isothermal experiments were carried out by Huang et al. [128] to develop the kinetic analysis of oxygen carrier reduction by char. A mixture of biomass char and $NiFe_2O_4$ as oxygen carrier was introduced into the TG-MS system and then heated to 1250 °C with different heating rates under the inert environment. The study investigated that the increase of activation energy of redox reaction corresponds to conversion ratio, and the random nucleation and subsequent growth model can be used to describe the kinetic mechanism. A new concept for syngas production was proposed to use a liquid metal oxide as an oxygen carrier for chemical looping gasification. The proposed configuration comprises two interconnected bubble reactors as the fuel and air reactors, and a liquid form of oxygen carrier is circulated between two reactors to provide the required heat and oxygen for the gasification reaction. The proposed concept could prevent the challenges from the solid oxygen carrier system such as agglomeration and sintering. Wu et al. [94] carried out a thermodynamic analysis to investigate the performance of four bimetallic oxygen carriers ($BaFe_2O_4$, $BaMn_2O_4$, $CaFe_2O_4$, $CaFe_2O_4$) for CLG of biomass-based lignin. The kinetic behavior and reaction performance of lignin pyrolysis and gasification was evaluated through a TGA-MS system. According to the results, $BaFe_2O_4$ and $CaFe_2O_4$ showed good syngas selectivity, while $BaFe_2O_4$ exhibited excellent reactivity and regeneration in the air after reduction.

## 5.2. Pilot-Scale Investigations

Along with thermodynamic analysis and kinetic study, BCLG in experimental test rigs has been intensively investigated in the temperature range of 600–1300 °C using vari-

ous types of biomass, typically woody biomass to evaluate the feasibility of the process. Different pilot-scale configurations have been used in the BCLG studies, commonly fixed bed [42,43,45,90] and fluidized bed reactors [25,26,32,33]. One of the key performances in those studies is the reactivity of oxygen carriers in the BCLG process. Various oxygen carriers have been investigated including Fe-based [26,32,43,67–70], Ni-based [25,44], Cu-based [71,72], Mn-based [28], Zn [73], bimetallic oxygen carriers (Fe-Ni [31,74], Fe-Cu [71,75], etc.). Additionally, the oxygen carriers were supported by supporting materials, such as $SiO_2$ [30], $Al_2O_3$ [26,33,42], CaO [43,85,86], MgO [45], $TiO_2$ [69,84], etc., to improve their reactivity and multifunctional properties, including catalytic function, tar decomposition and $CO_2$ adsorbent.

A 25 $kW_{th}$ system was developed by Ge et al. [30] to investigate the performance of rice husk gasification chemical looping using hematite as an oxygen carrier. The system mainly consists of a high-velocity fluidized bed as the AR and a bubble fluidized bed as the FR. The main components of hematite are $Fe_2O_3$, $SiO_2$, and $Al_2O_3$ with 83.21 wt.%, 7.06 wt.%, and 5.37 wt.%, respectively. The higher performance of BCLG with hematite, such as higher carbon could obtain at 860 °C, while carbon conversion efficiency reached a peak at oxygen carrier/biomass ratio of 1.0. Wei et al. [31] experimentally investigated the performance of Fe-Ni bimetallic oxides as oxygen carriers that were applied for biomass chemical looping gasification. The experiments were performed in a 10 $kW_{th}$ interconnected circulating fluidized bed which mainly consists of a fast fluidized bed as the AR and a bubbling bed as the FR. Sawdust of pine was used as feedstock. The experimental results indicated that the content of CO, $H_2$, and $CH_4$, as well as process performance, increased with the rising gasification temperature. The Fe-Ni bimetallic oxygen carriers displayed a higher gasification efficiency of biomass and showed a stable reactivity and good sintering resistance. Chuayboon et al. [73] proposed a novel process for chemical looping gasification. The high-temperature solar-driven chemical looping gasification of lignocellulosic with ZnO/Zn redox pair was developed to evaluate the advantages and reliability of the combined process. Experiments were carried out in a lab-scale solar reactor at a range temperature from 1050 to 1300 °C with a biomass/solid ratio from 0.5 to 1. Biomass feedstock was beech wood for the experiments. The experimental set-up mainly consists of the solar reactor, solar concentration system, gas injection, and particle feeding system, filtering unit, and gas analysis unit. A fixed ratio mixture of ZnO and biomass particles was placed in the hopper, and then the reactor was heated up to the required temperature by concentrated sunlight. During the study, the influence of temperature and reactant molar ratio on syngas production was analyzed. The result successfully demonstrated the feasibility of the process for the first time, and optimal operating conditions for the study were found at 1250 °C and a biomass/ZnO molar ratio of 0.75. Additionally, the temperature of Zn production was lower than that for ZnO thermal dissociation.

A study was carried out to evaluate the multi-functional oxygen carrier for hydrogen-rich syngas production [129]. An iron-based oxygen carrier was developed from the Bauxite residual of the alumina industry (red mud). It was evaluated the capacity of transport oxygen and heat from the air reactor to the fuel reactor to promote gasification, and catalytic function for fuel gasification and syngas reforming. Four series of experiments took place in a Bench-scale fluidized bed at 950 °C and atmospheric pressure to examine the performance of the process and oxygen carrier. Those experiments demonstrated that the oxygen carrier was able to promote endothermic gasification due to its stable transport capacity of oxygen and heat during multiple redox cycles. Furthermore, its reduced forms could strongly catalyze internal syngas reforming. Red mud oxygen carrier has a high melting point and proper heat capacity; thus, it can be operated at a high temperature in the FR.

Shen et al. [75] combined $Fe_2O_3$ and CuO to produce a novel oxygen carrier for biomass gasification. It was found that the bimetallic oxides showed advantages over two mono-metallic oxygen carriers in the investigation. A ratio of 50 wt.% of $Fe_2O_3$ and 10 wt.% CuO was selected as an optimal ratio in the study. Furthermore, the best per-

formance was observed at the O/C ratio of 0.75 and 900 °C. However, the combined oxygen carrier performed the worst reactivity after three cycles. It was mainly attributed to sintering caused by Cu atomic that resulted in the reactivity decline. During steam gasification of biomass, undesirable $CO_2$ and tar would be produced, which will reduce the quality of the hydrogen stream and process efficiency. The adding of calcium oxide (CaO) could be used to overcome such challenges. Calcium oxide could work as a tar reforming catalyst and $CO_2$ sorbent, and thereby increase hydrogen-rich gas production. However, the deactivation of calcium oxide after the carbonation reaction is a challenge for continuous hydrogen production and economic issues [91]. A study was conducted to evaluate the effects of Fe/Ca ratio on hydrogen production from chemical looping gasification of rice straw [86]. Synthesized bimetallic Fe-Ca oxides were used as an oxygen carrier in the study. Its results found that there are two types of calcium ferrites ($Ca_2Fe_2O_5$ and $CaFe_2O_4$) formed with different ratios of Fe/Ca; and at the Fe:Ca ratio of 1:1, the highest hydrogen yield was obtained with 23.07 mmol/g biomass at 800 °C. Furthermore, the process temperature needed for the completed redox of $Ca_2Fe_2O_5$ was higher than 800 °C during chemical looping gasification with steam.

Wu et al. [85] investigated CLG of biomass using steam and CaO additive. It found that CaO showed a positive effect on gasification performance, and worked mainly as a $CO_2$ absorbent at low temperature but as a catalyst above 700 °C. It was also found to retard the sintering and porosity reduction of $Fe_2O_3$. A novel process was proposed to combine BCLG and $CO_2$ splitting using $Ca_2Fe_2O_5$ aerogel as an oxygen carrier [90]. In the process, pine wood was decomposed with the presence of an oxygen carrier in the FR to produce syngas, phenolic-rich bio-oil, and biochar. The reduced oxygen carrier was re-oxidized in the AR by $CO_2$ instead of air to generate CO. The results showed that $Ca_2Fe_2O_5$ aerogel is a promising oxygen carrier due to its redox activity, phase reversibility and cyclic stability. The study also investigated the mechanism of the synergistic enhancement of chemical looping-based $CO_2$ splitting using biomass cascade. It was found that the products from biomass fast pyrolysis and oxygen carrier reduction can further benefit oxygen carrier reduction and biomass conversion, respectively, as well as enhance $CO_2$ reduction in the AR.

Continuous operations in a 1.5 $kW_{th}$ pilot were conducted for syngas production using pine wood as fuel and ilmenite [32], $Fe_2O_3/Al_2O_3$ [33] as oxygen carriers. A new method was proposed [32] for controlling the lattice oxygen used in the fuel reactor for syngas production by controlling the oxygen fed into the air reactor. The operating parameters in the BCLG in those studies were found that a high content of syngas components was produced during the process with 27–30% of $H_2$, 17–21% of CO for ilmenite and 37% of $H_2$, 21% of CO for $Fe_2O_3/Al_2O_3$ at autothermal conditions. It is noteworthy that tar generated in the studies was in the range of 0.9–3.0 g/$Nm^3$, being lower than that reported by other gasification technologies. Tian et al. [71] indicated that operating parameters strongly influence tar reforming in a BCLG process. The research results showed that tar decomposition is promoted at higher gasification temperatures, steam-to-biomass ratios and oxygen carrier-to-biomass ratios, resulting in the conversion of large molecular compounds into small ones. They also found that the presence of Cu in the oxygen carrier could enhance the decomposition of small molecular compounds in tar, while the Fe composition is favorable for a reduction in the yield of large molecular compounds in tar.

A novel oxygen carrier was synthesized for BCLG with additional functionalities of its reduced form acting as a catalyst for tar decomposition and a $CO_2$ adsorbent in syngas [130]. $Fe_2O_3$ is supported on silicalite-1 (Fe/S-1) was synthesized and characterized with different loadings as multi-functional materials to improve the quality of syngas and the efficiency of BCLG. The results found that the iron supported on silicalite-1 performed high thermal/chemical stability in a multi-cycle process and better catalytic activity in tar decomposition compared to the iron on a conventional silica support. An experimental investigation of CLG of torrefied woodchips using iron-based oxygen carriers was conducted in a bubbling fluidized bed reactor to evaluate the effects of operating parameters and the reactivity of oxygen carriers during the process [69]. The authors found that the operating

conditions strongly influence the process performance in terms of process efficiencies, gas yields. It was noted that the reactivity of iron-based oxygen carriers with different gaseous fuels follows the order $H_2 > CO > CH_4$. Additionally, the calcination temperature of oxygen carriers plays a key factor in the reactivity of oxygen carriers in the BCLG process. The other recent experimental studies of BCLG in the pilot-scale reactors are summarized and tabulated in Table 6.

**Table 6.** Summary of the key information of recent experimental studies in BCLG.

| References | Biomass | Type of OC | Fuel Reactor | Air Reactor | Remarks |
|---|---|---|---|---|---|
| Huijun et al. [25] Interconnected FB (25 kW$_{th}$) | Rice straw | NiO/A$_3$ | Bubbling fluidized bed Operating temperature: 650–850 °C | High-velocity fluidized bed | • Max. syngas yield (0.33 Nm3/kg) at 750 °C. <br> • Carbon conversion efficiency: max. value (60.28%) at SBR of 1.2 <br> • The addition of CaO improved Biomass gasification performance. |
| Huseyin et al. [26] Interconnected FB (10 kW$_{th}$) | Sawdust of Pine | Fe$_2$O$_3$/Al$_2$O$_3$ | Bubbling fluidized bed Operating temperature: 750–900 °C | Fast fluidized bed | • H$_2$ production was the highest value at 870 °C. <br> • Carbon conversion rate and gasification efficiency increased at higher temperatures. <br> • Oxygen carrier used has good stability and resistance to sintering |
| He et al. [34] | Biomass | NiFe$_2$O$_4$ | Fixed bed Operating temperature 700–900 °C | Fixed bed | • BCLG coupled with steam/CO$_2$ splitting. <br> • Producing syngas and pure H$_2$ or CO separately. |
| Yan et al. [42] | Sawdust | BaFe$_2$O$_4$/Al$_2$O$_3$ | Fixed bed reactor Operating temperature: 750–900 °C | - | • 30ABF showed high activity with char, but low reactivity with syngas. <br> • Max. syngas yield was at 850 °C, O/B of 0.6 and steam fraction of 33.6%. |
| Liu et al. [43] | Pine wood + polyethylene | CaO/Fe$_2$O$_3$ | Fixed bed reactor Operating temperature: 750–850 °C | Fixed bed reactor | • The optimal product was obtained at 850 °C and Fe$_2$O$_3$/feedstock of 0.25. <br> • The addition of 75% polyethylene increased H$_2$ yield to 1.59 Nm$^3$/kg. <br> • Cold gas efficiency and H$_2$/CO ratio were improved to 89.30% and 1.88. |

**Table 6.** *Cont.*

| References | Biomass | Type of OC | Fuel Reactor | Air Reactor | Remarks |
|---|---|---|---|---|---|
| Liu et al. [45] | Pine sawdust | $CaFe_2O_5$/ MgO, ZnO, $Al_2O_3$ | Fixed bed reactor Operating temperature: 850 °C | Fixed bed reactor | • The study evaluated the effects of Mg/Al/Zn oxides as support materials on $CaFe_2O_5$ reactivity in BCLG. <br> • Al oxide promoted the oxygen release rate by breaking the OC structure, but the syngas selectivity was reduced. <br> • ZnO showed higher reactivity in BCLG, but it was reduced into metal Zn before the reduction of the OC. <br> • MgO addition improved the oxygen release ability of the OC, as well as increased the performance of BCLG. |
| Zeng et al. [68] | Pine sawdust | Natural hematite ($Fe_2O_3$) | Fixed bed reactor Operating temperature: 800 °C | - | • The moisture content of biomass increased the gas yield to 1.1646 $Nm^3$/kg while steam increased the ratio of $H_2$/CO by about 16%. <br> • The diffusion of moisture content and steam showed opposite directions resulting in differing gasification reactivity. |
| Huang et al. [67] | Biomass char | Hematite ($Fe_2O_3$) | Fixed bed reactor Operating temperature: 850 °C | Fixed bed reactor | • Iron ore remained a good reactivity after 20 cycles. <br> • $Fe_2O_3$ was reduced into $Fe_3O_4$ and FeO under steam and inert atmosphere, respectively, during CLG of biomass char. |
| Hedayati et al. [28] | Woody based biomass | Mn ores | 300 W fluidized bed reactor 850–900 °C | Fluidized bed | • $C_3$ components were completely converted in all operating conditions. <br> • Syngas yield obtained was promoted by high fuel flow rate and high temperature. |

**Table 6.** *Cont.*

| References | Biomass | Type of OC | Fuel Reactor | Air Reactor | Remarks |
|---|---|---|---|---|---|
| Hildor et al. [131] | Wood pellet | Steel converter slag (LD slag) | Batch fluidized bed 820–970 °C | | • LD slag performed a high gasification rate/char conversion rate. <br> • LD slag acted as a catalytic for water gas shift reaction. <br> • Temperatures above 920 °C may increase the CO/C ratio and reduce the $H_2$/CO ratio. <br> • No $CO_2$ adsorption at temperatures above 800 °C due to carbonation. |

### 5.3. Simulation/Modeling Studies

Simulation and modeling of the CLG system are applied for prediction, evaluation, optimization. The most common configuration of chemical looping gasification mainly comprises two interconnected fluidized bed reactors and separators. Unlike the CLC process, minimal research has been conducted regarding the simulation and modeling of CLG by this time, particularly BCLG. The modeling of the system mainly includes its reaction kinetics, heat and mass transfer, and process performance. The most challenging issue is to describe the behavior of the gas-solid interactions inside the reactors. The model normally requires the configuration of the reactors, operational parameters, and the properties of gas and solids as input parameters to derive the combination of output parameters such as pressure and temperature distribution, composition and content of the stream, and energy and process efficiencies. The simulation of the gasification process can be developed based on [132]:

- Thermodynamic equilibrium
- Restricted thermodynamic equilibrium
- Kinetic mechanism
- Experimental data

Mathematical models of the CLG process available in the literature are mostly based on the computational fluid dynamic (CFD) technique. Currently, three main methods have been developed for the CFD modeling of fluidized-bed gasifiers such as the Eulerian-Eulerian approach, the Eulerian-Lagrangian approach, and the hybrid Eulerian-Lagrangian approach [133]. An alternative simulation approach for the gasification process is neural network modeling, but this method requires extensive experimental data, thus it is not often readily available [132]. An overall summary of simulation studies is tabulated in Table 7.

**Table 7.** Summary of the key information of recent simulation studies in BCLG.

| References | Biomass | Type of OC | Type of Model | Remarks |
|---|---|---|---|---|
| Gopaul et al. [134] | Poultry litter | CaO and $Fe_3O_4$ | ASPEN Plus (process modeling) | • The study compared the simulations of two CLG types for $H_2$ production.<br>• The iron-based OC model showed higher syngas yield obtained with 2.54 kmol/kmol biomass while the first model with CaO sorbent produced $H_2$-enriched syngas with 92.45 mol.%. |
| Detchusananard et al. [135] | Wood residue | NiO/CaO | ASPEN Plus (process modeling) | • CaO added played two roles of tar cracking catalyst and $CO_2$ sorbent.<br>• The simulation was developed based on the second-order respond surface model to analyze the process's energy efficiency performance.<br>• The maximum energy efficiency performance was obtained at S/C ratio of 2.6, 636 °C, CaO/C ratio of 1 and OC/C ratio of 1.06. |
| Cormos et al. [136] | Sawdust Coal/sawdust | Ilmenite | Computational methods | • The models were developed for the techno-economic evaluation of $H_2$ and power co-generation based on BCLG.<br>• The model used an ilmenite-based system to produce 400–500 MW net power with flexible $H_2$ output.<br>• The energy efficiencies obtained up to 42% with 99% of the carbon capture rate. |
| Aghabararnejad et al. [137] | Biomass | $Co_3O_4/Al_2O_3$ | ASPEN Plus (process modeling) | • A $7MW_{th}$ CLG model was developed in ASPEN Plus for techno-economic comparison with conventional systems.<br>• The CLG system had a higher total capital investment, but the annual operating cost was lower compared to other systems. |
| Li et al. [138] | Dried poplar | Iron-based ($Fe_2O_3$) | ASPEN Plus (process modeling) | • A BDCL process was developed with the three-reactor configuration based on a multistage model to simulate the performance of the process.<br>• The BDCL system showed a better performance compared to other conventional processes, and the feasibility of $CO_2$ capture. |
| Ge et al. [139] | Rice straw | Hematite ($Fe_2O_3$) | ASPEN Plus (process modeling) | • A CLG-BIGCC system was simulated using ASPEN Plus software.<br>• The model results showed a better performance with up to 4% higher compared to existing BIGCC plants in China.<br>• Five optimization schemes about CLG-BIGCC with nitrogen reinjection were proposed and investigated. |

**Table 7.** *Cont.*

| References | Biomass | Type of OC | Type of Model | Remarks |
|---|---|---|---|---|
| Kuo et al. [140] | Raw wood and torrefied wood | Iron-based | Matlab and ASPEN Plus | • A BSG-CLHP-CHP system was developed.<br>• The results showed the effects of operating conditions on the process performance.<br>• Hydrogen thermal efficiency and hydrogen yield of TW-derived syngas increased by 8.3% and 22%, respectively. |
| Li et al. [141] | Pine sawdust | Iron-based ($Fe_2O_3$) | Numerical method (CFD) | • The numerical method was developed to integrate the Eulerian multi-fluid model and the chemical reaction models to predict the concentrations of five gas components over time.<br>• The model is validated well by the experimental data from the literature.<br>• The results showed the continuous pine sawdust and mixing and segregation behaviors between fuel and OC particles strongly impact the concentrations of gas species and the evolution of the solid particle in the CLG system. |
| Li et al. [142] | Microalgae | $Fe_2O_3$ | Numerical method | • A one-dimensional transient model for BCLG was developed based on hydrodynamics and chemical reactions.<br>• The numerical model is to predict the time-varying outlet concentrations of gaseous components and process efficiencies.<br>• Energy and exergy analyses were carried out based on the validated model. |
| Dieringer et al. [17] | Wood pellet | Ilmenite | Matlab and ASPEN Plus (equilibrium process modeling) | • Two approaches for autothermal CLG behavior were analyzed.<br>• Evaluation of dilution of oxygen carrier with inert bed material.<br>• Optimizing process efficiency. |

Gopaul et al. [134] simulated two chemical looping gasification types using the AS-PEN Plus simulation software for $H_2$ production. The research aimed to compare the performance of two systems. The first CLG model used in situ $CO_2$ capture utilizing a CaO sorbent for the production of $CO_2$-rich stream for sequestration, and the second model used iron-based oxygen carriers in redox cycles to 99.8% of $Fe_3O_4$ recovery and higher syngas yields. The main findings of the model were syngas yields, gas content. The model showed that the second model produced more syngas yields, while the concentration of $H_2$ in the first model was higher than that of the second one. Furthermore, sensitivity analyses of temperature and pressure were carried out on the main factors to determine optimal operating conditions. A model was implemented by Detchusananard et al. [135] to investigate the sorption enhanced biomass chemical looping gasification without heating and cooling system for enriched hydrogen production. During the steam gasification, calcium oxide and solid oxygen carrier (NiO) were added as bed material. The model of gasification was developed using the ASPEN Plus process simulator to analyze the energy efficiency performance of the process. It is noticed that the energy efficiency depended on the steam to carbon molar ratio and gasifying temperature.

A large-scale biomass chemical looping for hydrogen and power co-production was assessed by Cormos [136]. This study aimed to evaluate the techno-economic feasibility of hydrogen and power co-generation based on the biomass direct chemical looping (BDCL) concept. The net power output of a plant model of about 400–500 MW$_{th}$ with a flexible hydrogen output in a range of 0 to 200 MW$_{th}$ (LHV) was developed for the assessment of the BDCL and the benchmark power plant concept. Computational methods (using ChemCAD software) based on mass and energy balances were used for in-depth techno-economic analysis. The influence of various technical and economic parameters on economics was considered through the sensitivity study. A 7-MW$_{th}$ CLG system was simulated to compare with conventional gasification processes using ASPEN Plus [137]. The CLG consists of a bubbling fluidized-bed gasifier and a fast fluidized-bed oxidized. Along with the comparison in the process performance, the economic aspects were also evaluated in the study. The results showed that the CLG system had a higher total capital investment compared to that of the conventional gasification process with pure oxygen but the annual operating cost of the CLG is $0.58M less resulting in the CLG as a feasible solution for biomass gasification.

Li et al. [138] developed a biomass direct chemical looping process (BDCL) for hydrogen and electricity co-production based on a multistage model using ASPEN Plus. The BDCL process configuration consists mainly of biomass preparation, the chemical looping system, heat recovery and steam generation (HRSG), gas cleanup units, and power generation systems. The BDCL process can produce hydrogen and/or electricity at any ratio, and it is 10–25% more efficient compared to conventional biomass combustion and gasification processes. The BDCL process also generated a high $CO_2$ concentration stream which could lead to a carbon-negative process from the life cycle standpoint. A system of biomass-based integrated gasification combined cycle coupling with CLG (CLG-BIGCC) for power generation was simulated by Ge et al. [139]. The system, mainly consisting of BCLG, gas cleaning, heat recovery steam generator (HRSG), and gas/steam turbine cycles, was developed with ASPEN Plus software. The simulation results were compared with the experimental data from a 25 kW$_{th}$ interconnected fluidized bed reactor under the same operating conditions. The power efficiency of the CLG-BIGCC system was 33.51%, the gas turbine efficiency obtained 33.24% and the efficiency of the steam turbine was 34.01%. Those efficiencies are higher than the existing BIGCC plants in China with a range of 30–32%. Moreover, a sensitivity analysis on the CLG-BIGCC was carried out to obtain the optimal gasification conditions, and five optimization schemes about CLG-BIGCC with nitrogen reinjection were proposed and investigated. Kuo et al. [140] developed a new system for the co-production of electricity and hydrogen with $CO_2$ capture using biomass as fuel. The model integrated a biomass steam gasification (BSG) with an iron-based chemical looping hydrogen production (CLHP) system and a combined heat and power (CHP) system. The fuel used in the study was raw wood (RW) and torrefied wood (TW). The results were used to evaluate the effects of operating conditions on the process performance and syngas derived from RW and TW. Using TW can improve the BSG-CLHP-CHP system's performance. Hydrogen thermal efficiency and hydrogen yield of TW-derived syngas increased by 8.3% and 22%, respectively.

Li et al. [141] developed a numerical model to investigate biomass gasification with iron-based oxygen carrier in a bubbling bed reactor. The model integrated the Eulerian multi-fluid model and the chemical reaction models including the decomposition of biomass, gasification of char, water-gas-shift reaction and the heterogeneous reactions between gases and metal oxides. It is validated well by the experimental data from the literature. The results were used to analyze the impacts of the mixing and segregation behaviors between two solid phases on the gas composition distribution at various operating conditions. A one-dimensional transient model, including the hydrodynamics and chemical reactions, was developed for biomass gasification using chemical looping technology to investigate the outlet concentrations of gaseous species and efficiencies at various operating conditions [142]. In the study, the numerical model was validated

against the experimental results utilizing $Fe_2O_3$ as oxygen carrier and microalgae as fuel. Additionally, the validated model also analyzed energy and exergy values to evaluate the energy sources. The analysis showed that the presence of oxygen carrier can improve the quantity and quality of syngas. [1,2]

## 6. Conclusions and Future Prospects

Biomass has been considered as a potential renewable source to replace fossil fuels for energy generation and chemical production. Due to the carbon cycle of biomass, the use of biomass as a feedstock can achieve significant environmental benefits of mitigation of net greenhouse gas emissions. However, biomass is currently less attractive in conventional thermochemical processes since it owns low energy density, high moisture content, complex ash composition, and highly distributed resource.

The comprehensive summary of biomass chemical looping gasification basis and its developments through recent studies will be of great benefit for the readers. This study is structured to investigate key components in a BCLG process. Therefore, through the existing knowledge and recent research, the potentials and challenges of BCLG are summarized in this study, as follows:

- Description of principles and developments of biomass-based chemical looping gasification technology.
- Recent research strategies and achievements in biomass-based chemical looping gasification.
- Prospects and challenges of biomass-based chemical looping gasification towards commercialization.

As discussed in the study, chemical looping technology can produce high-quality syngas using for energy generation or chemical production. Thus, using biomass as a feedstock for chemical looping gasification has many potential advantages over the conventional methods in biomass conversion and coal-fueled chemical looping processes. However, the chemical looping technologies are mostly in the R&D phase since they remain several challenges. Oxygen carrier is a key factor in the chemical looping concept, the selection of suitable oxygen carrier materials is one of the most important issues in terms of redox behavior, stability, availability, process performance, costs, environmental, and safety aspects. Additionally, tar formation from biomass chemical looping gasification has been considered increasingly. Tars formed during the process cause serious problems for the BCLG system, resulting in a decrease in the process efficiency. However, research on tar formation and elimination in BCLG is still limited. A better understanding of tars and finding efficient approaches to minimize their effects on the system are necessary for the commercialization of BCLG. Numerous studies on the chemical looping process of biomass have been carried out over the years, and many various carriers were investigated in a pilot-scale system of chemical looping gasification plants for their feasibility. The recent developments of biomass CLG have been carried out on a pilot-scale system ranging from a few kW to MW. This is the basis for the massive scale-up of the process that is necessary for commercial power plant and syngas production along with $CO_2$ capture. Furthermore, syngas cleanup technologies have been improving to mitigate problems regarding failure and blockage, as well as the loss of efficiency. Additionally, economic evaluation for each specific biomass feedstock, operational flexibility in the variation of feedstock composition is to respond to the changes in supply and demand of the feedstock market [9]. Those challenges should be solved to meet the coming future regarding the commercialization of this technology. Hence, chemical looping gasification using biomass as feedstock shows promising applications in the future in the continued fight against climate change [2].

**Author Contributions:** N.M.N. is responsible for administration, conceptualization, the original draft, and the applied methodology. F.A. supported the writing process with his review and edits. P.D. supported the writing process with his review and edits. B.E. supervised the research progress the presented work. All authors have read and agreed to the published version of the manuscript.

**Funding:** This work received no external funding. The corresponding author would like to thank the Techical University of Darmstadt, enabling the open-access publication of this paper.

**Institutional Review Board Statement:** Not applicable.

**Informed Consent Statement:** Not applicable.

**Data Availability Statement:** The data presented in this study are available in the article.

**Acknowledgments:** The authors would like to thank Technische Universität Darmstadt and the Institute for Energy Systems and Technology for support.

**Conflicts of Interest:** The authors declare no conflict of interest.

## Abbreviations

| | |
|---|---|
| AR | Air reactor |
| BCLG | Biomass chemical looping gasification |
| CFD | Computational fluid dynamics |
| CLC | Chemical looping combustion |
| CLG | Chemical looping gasification |
| CLR | Chemical looping reforming |
| CGC | Cold gas cleanup |
| CV | Calorific value |
| FR | Fuel reactor |
| GHG | Greenhouse gas |
| HHV | Higher heating value |
| LHV | Lower heating value |
| IEO | International energy outlook |
| LM | Looping material |
| HGC | Hot gas cleanup |
| OC | Oxygen carrier |
| WGC | Warm gas cleanup |
| TGA | Thermogravimetric analysis |

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
