# Peer review of "Biomass-Based Chemical Looping Gasification: Overview and Recent Developments"

_applsci, doi:10.3390/app11157069_

Round 1

Reviewer 1 Report

The paper is well written and gives an overview on the chemical looping gasification which is one of the under-developement techniques to enrich the syngas quality from biomass gasification. I added a very few comments on the PDF.

Author Response

Dear Reviewer #1

we appreciate the reviewer’s comments and we would like to thank Reviewer#1 again for his valuable suggestions. We hope the reviewer will find the detailed responses satisfactory.

Best regards,

The Authors

Author Response

Dear Reviewer #1

we appreciate the reviewer’s comments and we would like to thank Reviewer#2 again for his valuable suggestions. we tried our best to revise the manuscript. We hope the reviewer will find the detailed responses satisfactory.

Best regards,

The Authors

Reviewer 3 Report

no

Author Response

Dear Reviewer #3

We appreciate the reviewer’s time to review our manuscript. We tried out best to revise the manuscript. We hope the reviewer will find the detailed responses satisfactory.

Best regards,

The Authors

Reviewer 4 Report

The paper is well organized and the the thematics well argumented too. I found the review very interesting and very easy to read. All concerns about the  CLBG technologies were treated with sufficient accuracy to well introduce the reader in the field. Again the paper offers a lot of ideas for a deeper understanding of the various arguments. The current state of art about the tecnologies is well presented as well as current research activities. It is my opinion that the work could be accepted after minor revision. Only few comments:

Pag. 6 :

Additionally, since biomass has higher reactivity and higher volatile content, biomass gasification can occur at lower temperatures that reduce the extent of heat loss, emissions, and material problems associated with high temperatures.

It is not properly true, because at low temperature, very high tar content in syngas is produced. Process temperature is surely lower if compared with combustion but not enough to prevent some issues like that with materials. The reference temperature is in the range 800 - 900 °C.

Pag. 7:

However, the drawback of conventional gasification technology is the demand for pure oxygen as a gasifying agent to produce high-quality syngas and a large amount of heat supply, making the process less attractive.

In gasification processes conventionally, air is used as gasifying agent. This surely leads to a low LHV syngas but it is economically more feasible if compared with other solutions. If steam is used in addition to air, a more high hydrogen content syngas can be produced. Again, if you consider oxygen or steam/oxygen (or more properly enriched air - i.e. up to 40-50 % vol. O2 -) a very high LHV syngas is produced. This latter configuration requires proper plant setup, and related costs are very very high so that it is pratically not feasible. 

Pag. 8:

Since most gasification reactions are endothermic, ..

but also reduction reaction of metal oxide.

Pag. 18:

Drescher et al. [43] proved that a small biomass power plant with Organic Rankine Cycle has higher efficiency at low temperatures; therefore, An Organic Rankine Cycle could be applied for biomass chemical looing gasification.

Do you mean biomass gasification power plant or combustor?

How do you think to integrate ORC in BCLG power plant?

Pag. 32:

Additionally, biomass ash has considerably low melting temperatures (< 800 °C), …

In my opinion this statement isn't properly true. Surely, the low melting temperature of ashes is a real issue when fluidized beds were used, but it must be consedered that for most common biomass used in gasification processes, this temperature is above 900 - 1000°C.  Agglomerations can also occur in these cases because of hot spot inside the reacting bed.

Author Response

Dear Reviewer #4

we appreciate the reviewer’s comments and we would like to thank Reviewer#4 again for his valuable suggestions. We hope the reviewer will find the detailed responses satisfactory.

Best regards,

The Authors

Round 2

Reviewer 2 Report

The manuscript has significant improvements.  It is a valuable, in-depth, and informative review. 

Only suggestions are fixing missing references/errors on pages where "Error!..." is shown (examples are on page 1 Line 17, page 2 Line 41, page 6 Line 234, page 7 Line 244, etc.).